# Sensitivity of Arctic sulfate aerosol and clouds to changes in future surface seawater dimethylsulfide concentrations

Rashed Mahmood[1,2], Knut von Salzen[1,2], Ann-Lise Norman[3], Martí Galí[4] and Maurice Levasseur[5]

[1]School of Earth and Ocean Sciences, University of Victoria, Victoria, British Columbia, Canada
[2]Canadian Center for Climate Modelling and Analysis, Environment and Climate Change Canada, Victoria, British Columbia, Canada
[3]Department of Physics and Astronomy, University of Calgary, Calgary, Alberta, Canada
[4]Takuvik Joint International Laboratory & Québec-Océan, Université Laval, Québec, Québec, Canada
[5]Département de biologie and Québec-Océan, Université Laval, Québec, Québec, Canada

*Correspondence to* : Knut von Salzen (knut.vonsalzen@canada.ca)

**Abstract.** Dimethylsulfide (DMS), outgassed from ocean waters, plays an important role in the climate system, as it oxidizes to methane sulfonic acid (MSA) and sulfur dioxide ($SO_2$), which can lead to the formation of sulfate aerosol. Newly formed sulfate aerosol resulting from DMS oxidation may grow by condensation of gases, in-cloud oxidation, and coagulation to sizes where they may act as cloud condensation nuclei (CCN) and influence cloud properties. Under future global warming conditions, sea-ice in the Arctic region is expected to decline significantly, which may lead to increased emissions of DMS from the open ocean and changes in cloud regimes. In this study we evaluate impacts of DMS on Arctic sulfate aerosol budget, changes in cloud droplet number concentration (CDNC), and cloud radiative forcing in the Arctic region under current and future sea ice conditions using an atmospheric global climate model. Given that future DMS concentrations are highly uncertain, several simulations with different surface seawater DMS concentrations and spatial distributions in the Arctic were performed in order to determine the sensitivity of sulfate aerosol budgets, CDNC, and cloud radiative forcing to Arctic surface seawater DMS concentrations. For any given amount and distribution of Arctic surface seawater DMS, similar amounts of sulfate are produced by oxidation of DMS in 2000 and 2050 despite large increases in DMS emission in the latter period due to sea ice retreat in the simulations. This relatively low sensitivity of sulfate burden is related to enhanced sulfate wet removal by precipitation in 2050. However simulated aerosol nucleation rates are higher in 2050, which results in an overall increase in CDNC and substantially more negative cloud radiative forcing. Thus potential future reductions in sea ice extent may cause cloud albedos to increase, resulting in a negative climate feedback on radiative forcing in the Arctic associated with ocean DMS emissions.

## 1 Introduction

Dimethylsulfide is produced in the surface ocean by biological processes that involve phytoplankton, zooplankton, and bacteria (Simó 2001; Stefels et al. 2007). A fraction of the surface seawater DMS is vented to the atmosphere, depending on turbulence at the air-water interface (generally parameterized as a function of wind speed; Wanninkhof et al., 2009; see also Jahne et al., 1987; Merlivat and Memery, 1983; Memery and Merlivat, 1985; Monahan and Spillane, 1984) and

the depth of the oceanic upper mixed layer that exchanges with the atmosphere (Galí and Simó, 2015). Atmospheric DMS is subsequently oxidized to MSA and $SO_2$. The latter is further oxidized to sulfuric acid ($H_2SO_4$), which can cause formation of new aerosols or condense on pre-existing aerosols. These aerosols may then act as CCN and affect cloud microphysical properties, especially in remote marine environments where concentrations of other types of CCN are low (Clarke et al., 1998; Leaitch et al., 2013; Dall'Osto et al., 2017, Collins et al., 2017). Willis et al. (2016) found that gaseous MSA may also play an important role in the initial growth of new particle formation.

According to the so-called CLAW hypothesis (Charlson et al., 1987), a negative feedback loop operates between ocean ecosystems and the Earth's climate. In particular, formation of new aerosol particles from ocean DMS emissions to the atmosphere leads to increased cloud albedo and reduced surface ocean temperature and/or incident irradiance, which then suppresses production of DMS in the ocean and emission to the atmosphere. Recent studies concluded that there is little evidence to support CLAW under present day climate conditions (Woodhouse et al., 2010; Quinn and Bates, 2011; Browse et al., 2014) since the sources of cloud condensation nuclei in marine boundary layer are numerous and the response of clouds to changes in aerosols are more complex than previously thought at the time of CLAW hypothesis (Quinn and Bates, 2011). However, marine DMS emission may still drive submicron aerosol populations over much of the remote marine atmosphere (Quinn et al., 2017). Important local impacts of DMS on climate may still exist in the Arctic where summertime aerosol and clouds are strongly influenced by DMS (Leaitch et al., 2013). Newly formed particles influence cloud albedo through cloud microphysical processes and thus can influence local radiation budgets and feedback mechanisms. Considerable concentrations of ultrafine particles and DMS, which are likely involved in naturally occurring aerosol/climate interactions, have been observed in the Arctic in summer (Willis et al., 2016; Ghahremaninezhad et al., 2016; Burkart et al., 2017). Furthermore, Grandey and Wang (2015) found that artificially enhanced DMS emissions in different latitude bands could potentially offset greenhouse gas induced warming across most of the world and especially in the Arctic region. The loss of Arctic sea-ice allows further penetration of sunlight into surface ocean water (e.g. Nicolaus et al., 2012) that can increase net production of algae and phytoplankton (e.g. Arrigo and van Dijken, 2015). In addition, when sea-ice is melted the surface ocean water is more prone to wind stress (Rainville et al., 2011; Martin et al., 2014) that can enhance air-fluxes of DMS and other gases (e.g. Bates et al. 2006). Possible increases in DMS emissions and sulfate aerosol concentrations are particularly important in the Arctic where changes in aerosol radiative forcings are amplified by powerful Arctic feedback processes, including the surface albedo feedback (Pithan and Mauritsen, 2014; Gagné et al. 2015).

Arguably, uncertainties in surface seawater DMS concentrations and parameterizations of DMS emission fluxes limit scientific progress on climate effects associated with CLAW. For instance, the widely used climatology by Lana et al. (2011) is based on a compilation of data sets from different Arctic field campaigns that took place during the time period from 1985 to 2008. Only measurements from the warm season were used, mainly from the Atlantic portion of the Arctic. The small number of measurements from other locations in the Arctic is problematic, as recent research in the NETCARE network (Abbatt et al., 2019) has shown. Surface seawater DMS concentrations measured in the Canadian Arctic in July and August of 2014 and 2016 were substantially higher than those used by Lana et al. for July and August

(e.g., NETCARE median concentrations of 4.4 nmol/L and 7.3 nmol/L, Martine Lizotte, personal communication; median concentration range from 0.5 to 4.4 nmol/L for Lana et al., https://saga.pmel.noaa.gov/dms/). Furthermore, melt ponds on sea ice represent a yet missing source of DMS in studies of the Arctic (Mungall et al., 2016; Ghahremaninezhad et al., 2016; Gourdal et al. 2018; Abbatt et al., 2019). Galí et al. (2018) argue that biases in the climatology by Lana et al. arise from the application of objective interpolation procedures to a limited amount of

measurements. Consequently, Arctic surface seawater DMS concentrations based on Lana et al. (2011) differ substantially from those of an earlier climatology (Kettle and Andreae, 2000), ocean biogeochemical models, and DMS parameterizations (Tesdal et al., 2016a), indicating large uncertainties in estimates of surface seawater DMS concentrations.

Over the last few decades, Arctic temperature has increased at a rate much faster than in other parts of the world (ACIA, 2005; AMAP, 2017). Enhanced Arctic warming is largely caused by sea-ice-albedo feedbacks and it is expected that Arctic summer sea-ice may completely disappear well before the end of this century if warming continues at rates simulated by current climate models (Stroeve et al., 2012). Browse et al. (2014) found a weak response of CCN concentrations to enhanced Arctic DMS emissions from complete loss in summer sea-ice due to efficient scavenging of

aerosol by drizzle associated with stratocumulus clouds. They did not find evidence for climate feedbacks through changes in cloud properties from enhanced aerosol sources in an ice-free summertime Arctic. However, Browse et al. used an atmospheric chemical transport model with specified meteorological conditions, which excludes responses of Arctic clouds and precipitation to changes in sea ice conditions. In a subsequent study, Ridley et al. (2016) performed fully interactive simulations of sea-ice, ocean biology (with DMS estimated using an embedded empirical algorithm;

Simó and Dachs, 2002), aerosols, and clouds with the HadGEM2-ES model. They found a two- to five-fold increase in DMS emissions, an increase in sulfate CCN concentration, and an associated 1 W m$^{-2}$ reduction in simulated summer cloud shortwave radiative forcing in the Arctic.

      Despite substantial research activities (e.g. Gabric et al., 2005, Thomas et al., 2010, Woodhouse et al., 2010; Browse et

al., 2014; Ridley et al., 2016) it is still very challenging to estimate DMS emissions and even more so how sea-ice reductions may affect DMS emissions in the Arctic in the future. Results of several modeling studies indicated that global warming related sea-ice loss would result in enhanced DMS emission fluxes in the Arctic region (Bopp et al., 2003; Gabric et al., 2005; Levasseur, 2013; Browse et al., 2014; Galí et al., *submitted*). Other studies suggest that the changes in DMS emission flux may be negative in sign due to potentially enhanced ocean acidification (e.g. Six et al.,

2013; Schwinger et al., 2017). Phytoplankton species composition is a controlling factor for DMS concentrations given the wide range of cellular dimethylsulfoniopropionate (DMSP) quota among different phytoplankton species (Stefels et al. 2007). For example, the cellular DMSP quota of haptophytes is greater than that of diatoms by a factor of >10 (Stefels et al. 2007). Long-term observational studies provide evidence that high DMSP-producing haptophytes are becoming more prevalent in the Arctic in the last decade (Winter et al., 2013; Nöthig et al., 2015; Soltwedel et al., 2016).

Furthermore, Arrigo et al. (2008) suggest that primary productivity may increase more than 3 times compared to 1998-2002, if Arctic sea ice loss continues. A combination of a shift in the species composition and an increase in primary

productivity (e.g. Yool et al., 2005; Vancoppenolle et al., 2013) could imply a multiplicative increase in surface seawater DMS concentrations in future climate.

Given large uncertainties in present-day surface seawater DMS concentrations and potentially large increases in future concentrations, the approach in the current study is to consider a wide range in concentrations in the Arctic for present-day and future conditions. The lower bound of the concentration range is provided by Lana et al. (2011), and the upper bound is obtained by scaling these concentrations by a factor of 10. A number of scenarios for DMS concentrations within this range are considered. These will be used to determine relationships between surface seawater DMS
concentrations and climate variables in the Arctic. By selecting widely different scenarios, the robustness of the relationships can be tested without making a priori assumptions that only apply to specific DMS scenarios. Model sensitivity tests with similarly enhanced DMS concentrations have previously been performed (e.g. Grandy and Wang, 2015; Fiddes et al., 2018). Sensitivities of sulfate aerosols, CDNC and cloud radiative forcing to different DMS emission scenarios are investigated using a state-of-the-art atmospheric global climate model. Based on the sensitivity simulations
we provide an assessment of Arctic annual mean changes in sulfate aerosol budget and cloud microphysical properties in relation to the mean DMS concentration in the Arctic. Note that an evaluation of surface seawater DMS data sets is outside the scope of the study.

## 2 Summary of model features


We used version 4.3 of the Canadian Atmospheric Model (CanAM4.3) which is an improved version of CanAM4 (von Salzen et al., 2013). The improvements to version 4.3 compared to version 4 include a higher vertical resolution, improved parameterizations for land surface and snow processes, DMS emissions, and clear-sky radiative transfer. CanAM4.3 has 49 vertical levels extending up to 1 hPa with a resolution of approximately 100 meters near the surface.
Model simulations are performed using a spectral resolution of T63 which is equivalent to the horizontal resolution of approximately $2.8 \times 2.8$ degrees. The model uses separate parameterizations for layer and convective clouds. Aerosol microphysical processes are based on the Piecewise Lognormal Approximation (von Salzen, 2006; Ma et al., 2008; Peng et al., 2012; Mahmood et al., 2016; AMAP, 2015). The model simulates binary homogeneous nucleation of sulfuric acid and water vapour. Newly formed particles grow by condensation and coagulation. The numerical treatment of these
processes is highly accurate and compares well with other methods (von Salzen, 2006). A detailed description of parameterizations of ocean DMS flux to atmosphere, oxidation and removal processes is provided in Tesdal et al. (2016a). Briefly, surface seawater DMS is ventilated to the atmosphere based on modeled wind speed and the piston velocity parameterization of Nightingale et al. (2000). There are no DMS emissions from sea ice. Furthermore, the model does not account for emissions of organic aerosol species from the ocean.


In the atmosphere, DMS is oxidized to MSA and $SO_2$ by hydroxyl (OH) radicals during day time and nitrate radical ($NO_3$) during night, with further oxidation of $SO_2$ to sulfuric acid ($H_2SO_4$) by OH in clear-sky conditions. MSA is treated as sulfuric acid in model for simplicity. Binary homogeneous nucleation of $H_2SO_4$ and water vapour may cause formation of new aerosol particles, depending on temperature and relative humidity (Kulmala et al., 1998; von Salzen et

al., 2000). In-cloud production of sulfate requires ozone ($O_3$) and hydrogen peroxide ($H_2O_2$) as oxidants (von Salzen et al., 2000), with oxidant (OH, $NO_3$, $H_2O_2$, $O_3$) concentrations specified according as climatological results from the Model for Ozone and Related Chemical Tracers (MOZART, Brasseur et al., 1998). Dry deposition of aerosol depends on concentrations of aerosols in the near surface model layer (Zhang et al., 2001). Wet deposition includes in-cloud scavenging in both convective clouds and layer clouds, and below-cloud scavenging. Emissions of non-DMS biogenic and anthropogenic aerosol precursors and primary aerosols for the time period up to year 2000 are specified according to Lamarque et al. (2010) and after that according to the IPCC RCP4.5 scenario (Moss et al., 2010).

Cloud droplet number concentrations are calculated based on the assumption of a parcel of air which ascends from the subcloud layer into the cloud layer with a characteristic vertical velocity (Peng et al., 2005), where the standard deviation of the subgrid-scale cloud vertical velocity probability distribution is parameterized using the approach by Ghan et al. (1997). Aerosol particles that are suspended in the parcel of air may activate and grow into cloud droplets by condensation of water vapour. A numerically efficient solution of the condensational droplet growth equation (e.g. Seinfeld and Pandis, 1998) is employed for this purpose. In grid cells that are affected by clouds, CanAM4.3 accounts for cloud albedo and lifetime effects (first and second aerosol indirect effects) as well the semi-direct effect. Parameterizations of droplet evaporation in the model do not account for aerosol effects, similar to Coupled Model Intercomparison Project phase 5 (CMIP5) climate models. Although aerosol indirect effects are very difficult to constrain, some studies based on observations and cloud-resolving modelling indicate that cloud microphysical processes may produce negative or positive radiative forcings, depending on the meteorological situation and nature of the clouds (Stevens and Feingold, 2009).

**3 Sulfate concentrations in CanAM4.3**

We performed two sets of historical model simulations, one with the full model with all natural and anthropogenic aerosols and their precursors included ("hisCont") and one with zero surface seawater DMS ("hisNoDMS"). Monthly mean surface seawater DMS concentrations in hisCont are specified according to the climatology of Lana et al. (2011) [hereafter referred to as L10]. The global annual DMS emission flux in hisCont is 24.96 TgS/yr, which is very close to 25.3 TgS/yr reported in Tesdal et al. (2016a) and well within previously reported ranges (e.g. Lana et al., 2011; Tesdal et al., 2016a). Both simulations were integrated for the time period 1991 to 2003 during which extensive observations of sulfate are available. Wind and temperature in each simulation were nudged towards specified results from a common simulation with CanAM4.3 for this time period. According to Kooperman et al. (2012) nudging of meteorological model variables reduces the influence of natural variability and therefore improves estimates of differences in diagnosed aerosol indirect effects. Similarly, biases in simulated aerosol and CDNC concentrations between the simulations are also reduced according to a statistical analysis of CanAM model results (not shown). The contribution of DMS oxidation to total sulfate concentrations is determined by calculating the difference in simulated sulfate concentrations between these two simulations (i.e. hisCont - hisNoDMS), which is interpreted as biogenic sulfate in the following.

A slightly different version of CanAM has previously been evaluated (Eckhardt et al., 2015; Tesdal et al., 2016b). In a multi-model comparison, Eckhardt et al. (2015) compared model simulations of sulfate and black carbon aerosols with observations from different stations and aircraft campaigns and found that most models, except CanAM, significantly underestimated observed concentrations in the Arctic region. Mahmood et al. (2016) found that black carbon concentration differences in four models are related to differences in wet removal processes in the models.

For the current study we used observed data from various ship-based campaigns and observations from Alert in Canada to further validate simulations of sulfate, with particular emphasis on the role of biogenic emissions. Shipboard data from the National Oceanic and Atmospheric Administration, Pacific Marine Environmental Laboratory (NOAA PMEL) was obtained from cruises that fell within the period 1992-2002. Only non-sea-salt sulfate (nss-$SO_4^{2-}$) data was selected and summed for all available bin sizes. The gridded model data was matched to the nearest location of the observations, shown in Fig. 1a.

Figure 1 shows that simulated sulfate concentrations agree well with observations, especially in regions where modeled DMS contributions are relatively large. The mean value for all ship-based observations is 3.416±4.018 (µg/m³) and the model mean value is 2.079±1.815 (µg/m³), corresponding to a model underestimate of ~39%. Most of the underestimates in simulated mean concentrations are associated with locations where the model simulates a large contribution of fossil-fuel (non-biogenic) sulfate to total sulfate concentrations.

From the Alert data, with highly variable contributions of DMS, it is evident that the model overestimates the contribution of DMS to total sulfate concentrations at this location (Fig. 1b). Overall, CanAM4.3 is able to capture the sulfate annual cycle very well at Alert, with slight underestimation in winter and spring and overestimation in summer (Fig. 2). The correlation coefficient for the mean annual cycle between model and observations is 0.95. Mean observed and simulated sulfate concentrations at Alert are 0.475±0.413 and 0.419±0.228 µg/m³ respectively, corresponding to a model underestimate of ~12%.

An analysis of isotopic data is available for Alert, which can be used to distinguish between contributions of biogenic and fossil-fuel sources to sulfate concentrations in the observations at this site (Norman et al., 1999). The ratio between sulfur isotopes ³⁴S and ³²S of an observed sample is compared with an international standard ratio based on sulfur isotopes in Vienna-Cañon Diablo Troilite (Beaudoin et al., 1994; Krouse and Grinenko, 1991; Norman et al., 1999). The results are expressed in parts per thousand (‰). The sulfate concentration at Alert consists of the sum of marine biogenic, anthropogenic, and sea salt sulfate with the delta isotopic ratios of +17.5‰, +5‰, +21‰ respectively.

A comparison with the isotopic data indicates that the model underestimates fossil-fuel sulfate and overestimates biogenic sulfate (Fig. 2b and 2c). This difference is particularly pronounced in spring and early summer, when observed biogenic sulfate concentrations are particularly high. An interesting feature is the double peak in biogenic sulfate concentrations during the annual cycle with one peak occurring in May and the other in October (Fig. 2c), which is essentially captured by the model. The occurrence of October peak at Alert has not been investigated. Sharma et al.

(2012) showed that MSA concentrations at Alert are anti-correlated with sea ice fraction. It is possible that the peak in October is related to increased fluxes of DMS into the Arctic atmosphere due to the minimum in sea ice fraction in September. It is also possible that DMS is transported to Alert from lower (subpolar) latitudes, where fall phytoplankton blooms are a dominant feature of the marine ecosystem. The correlation coefficient for the mean annual cycle between observed and simulated biogenic sulfate concentrations is 0.73.


Another interesting feature is the relative contribution of biogenic sources to total sulfate concentrations (Fig. 2d). Although absolute sulfate aerosol concentrations in summer are much lower than during other seasons, both observations and model results indicate a much larger contribution of biogenic sources to total sulfate concentration in this season compared to other seasons (Fig. 2d).


**4 Sensitivity of sulfate, clouds, and radiation to changes in DMS**

Five ensembles of five simulations each were performed, where ensemble members were generated by introducing random perturbations in radiative flux calculations (a total of 25 simulations). The five experiments differ in terms of
specified surface seawater DMS concentrations in the Arctic region, defined here as the region from 62.78˚-90˚N. A wide range of different Arctic surface seawater DMS concentration patterns is considered in order to account for substantial uncertainties in surface seawater DMS concentrations. For present day, uncertainty in specified Arctic DMS concentration climatologies arises from a lack of observational data, and concentrations that are highly variable in space and time (Lana et al., 2011; Tesdal et al., 2016a; Galí et al., 2018). Furthermore, very little is known about how DMS
concentration may evolve in the future. Outside the Arctic, monthly mean ocean DMS concentrations in the simulations are specified according to the L10 climatology.

For the first set of simulations, DMS concentrations were specified as zero in the Arctic, hereafter referred to as CNTRL. All subsequent simulations are compared to CNTRL in order to estimate the contribution of Arctic DMS emissions to
simulated biogenic sulfate concentrations and climate. For the second set of simulations, CLIM, the L10 monthly climatology is used in the Arctic. In 10×CLIM, the L10 climatology was multiplied by a factor of 10 at each model grid point in the Arctic region. In order to analyze the impact of spatial variability in DMS concentrations, a single DMS concentration (i.e. 16.9 nM) was assigned to each grid cell in the Arctic region hereafter referred to as UNFM. The number used for DMS concentration in UNFM is arbitrarily calculated as the grid-point average of 10 times the L10
climatology in the Arctic. It provides an additional scenario for testing the sensitivity of forcings under uniform surface seawater concentrations of DMS and also provides a test of linearity of such responses. Finally, as a further test of spatial variability of DMS, we used a satellite-based estimate of surface seawater DMS concentration (Galí et al., 2018) and multiplied it by a factor of 10 in 10×SAT. A value of 5 nM was applied in 10xSAT over the central Arctic region where satellite observations are not available, upon the observation that available sea-surface DMS measurements in the
Arctic winter have an average of ~0.5 nM. Note this earlier version of Galí et al. (2018) satellite based DMS estimation had a small negative bias in magnitude, however, the spatial distribution remained largely unchanged after correction. Surface seawater DMS concentrations in all simulations are summarized in Fig. 3 and Table 1.

Simulated horizontal wind and temperature in each individual member of an ensemble (i.e. 5 separate simulations) were nudged towards specified results from a corresponding simulation (i.e. separate free running model simulation) with CanAM4.3 using a nudging time scale of 6 h. The model was integrated over four years for 1998-2001 and annual mean model results during the last three years of the simulations were analyzed (hereafter referred to as 2000). Present-day sea ice amounts and sea surface temperatures (SSTs) are specified according to reanalysis data from Climate Forecast System Version 2 (Saha et al., 2014). In addition, all of the above experiments were repeated for 2048-2051 (referred to as 2050). The projected sea ice amounts and SSTs from a 50-member ensemble of simulations with CanESM2 for the RCP8.5 emission scenario were used in order to represent conditions in 2050 (Sigmond and Fyfe, 2016). Given the very large size of this ensemble, impacts of simulated natural variability on mean simulated sea ice and sea surface temperatures are negligible (Sigmond and Fyfe, 2016). The exact same greenhouse gas concentrations and emissions are specified for each individual ensemble member according to the RCP8.5 scenario. In order to further minimize the impact of natural variability in atmospheric and aerosol microphysical processes in simulations with CanAM4.3, we use mean results from 5 different CanAM4.3 ensemble members with the exact same boundary conditions and baseline emissions. Ensemble members were generated by introducing random perturbations in radiative flux calculations which leads to small differences in meteorological conditions for each ensemble member. Similar to the approach used in comparable aerosol modelling studies using CMIP5 data (e.g. Ekman, 2014), our method ensures that ensemble mean results are robust and consistent with the boundary conditions and emissions that were used in the simulations. In CanESM2, the Arctic is devoid of sea ice in September by 2050, consistent with results from other models for this relatively high emission scenario (Stroeve et al., 2012). The difference of sea ice extent in the simulation time periods is summarized in Fig. S1 (in supplementary data). The Arctic annual mean sea ice fractions are 75.6% (2000) and 50% (2050) for grid cells where the sea ice fraction is 0.15 or larger. Similar to simulations corresponding to year 2000, simulated horizontal winds and temperature were nudged towards specified results from 5 simulations with different meteorological conditions.

The total annual Arctic DMS fluxes for the two simulations time periods are summarized in Table 1. For 2000, DMS emission fluxes are approximately linearly proportional to the mean surface seawater DMS concentrations. For example, the total annual Arctic fluxes for the 10xCLIM are 84% higher than for 10xSAT, corresponding to 76% higher concentrations of Arctic-mean surface seawater DMS. This indicates that the sensitivity of Arctic-mean DMS fluxes to deviations of spatial distributions from the mean surface seawater concentrations is relatively low for 2000. Similar results for 2050 give evidence for relatively low sensitivity of fluxes to spatial distributions of surface seawater DMS concentrations.

Sea ice fraction in 2050 in summer and autumn is much lower than in 2000 (Fig. S1). For CLIM and 10xCLIM the total Arctic sulfur flux increases by 33% from the earlier to the later time period due to the reduction in future sea ice fraction. The difference is up to 47% for 10xSAT and 53% for UNFM (Table S1). Regionally, differences in fluxes are strongly correlated with changing sea ice fractions, with increases in regions with reduced sea ice fraction in 2050 and only minor changes over the open ocean (Fig. 4).

Owing to large DMS emission fluxes in the Atlantic region, the spatial pattern of the biogenic sulfate burden (relative to CNTRL) produces a maximum in this region for all surface seawater DMS data sets (Fig. 5). However, for CLIM diagnosed biogenic sulfate burdens are statistically significant over the Greenland Sea and nearby Baffin Bay for 2050 but not for 2000. Differences in Arctic-mean biogenic sulfate burdens between 2000 and 2050 are relatively small for all of the scenarios, ranging from just -1% for UNFM to +21% for CLIM, despite the relatively large increases in DMS emissions between 2000 and 2050 (Table 1 and Fig. 5). The weak responses in biogenic sulfate burdens to DMS emissions are caused by increased precipitation and aerosol wet removal in the Arctic in 2050 (Tables S1 and S2; Figs. S2 and S3). Thus the wet deposition of biogenic sulfate from Arctic DMS emissions becomes more efficient in the future. Whereas emissions of Arctic DMS increase between 33.3% (CLIM) and 53.2% (UNFM), wet deposition of biogenic sulfate from Arctic DMS emissions increases more strongly, between 42.45% (CLIM) and 72.1% (UNFM) from 2000 to 2050 (Table S2). The fraction of Arctic DMS emissions that is removed by wet deposition increases from between 55.3% (UNFM) and 76.5% (CLIM) in 2000 to between 62.1% (UNFM) and 81.8% (CLIM) in 2050 (Table S3).

On the other hand, projected reductions in anthropogenic sulfur emissions between 2000 and 2050 lead to reductions in total wet deposition of sulfate in the Arctic by -47.7% in CNTRL (Table S1). In the sensitivity experiments with increases in Arctic emissions of DMS between 2000 and 2050 reductions in total sulfate wet deposition in the Arctic between 2000 and 2050 are weaker, i.e. between -7.5% (10×CLIM) and -40.6% (CLIM). Considering the very wide range of surface seawater DMS concentrations applied here, a nearly complete compensation of aerosol production from oceanic DMS by increased wet deposition seems to be a robust feature of the future Arctic, largely independent of DMS emission patterns and amounts.

Changes in CDNC are important for radiative effects of sulfate aerosols. Impacts of climatological DMS emissions on CDNC are not statistically significant in CLIM, i.e. they are within the range of meteorological variability in the ensemble of simulations (Fig. 6). The relatively weak simulated impact of present-day climatological DMS concentrations on CDNC and cloud microphysics is in agreement with previous studies (e.g. Browse et al., 2014; Ridley et al., 2016). Similarly, for 2050, few regions in the Arctic show significant impacts of present-day DMS emissions on CDNC although local increases are up to about 10%. On the other hand, the other sets of simulations (i.e. 10×CLIM, UNFM and 10×SAT) produce significant changes in CDNC, especially for 2050, with increases up to ~$10^7$ m$^{-3}$ for 10×CLIM. It is interesting to note that although the biogenic sulfate burdens are similar in 2000 and 2050, there are relatively large systematic increases in CDNC due to increased Arctic DMS emissions in 2050 for these simulations.

Increases in CDNC between 2000 and 2050 are related to increases in formation of new particles in the lower Arctic troposphere by between +128 and +269% (Table 2 and Fig. S3) for the range in surface seawater DMS concentration considered. This leads to large-scale increases in CCN concentrations near the surface in the Arctic (Fig. S3), which is in contrast to a more non-uniform response of CCN concentrations to reductions in sea ice fraction according to Browse et al. (2014), with relatively large simulated increases over the continental Arctic and small decreases over the central Arctic Ocean. Large-scale increases of CCN concentrations and nucleation rates in 2050 in simulations with CanAM4.3

can be attributed to several factors: First, global anthropogenic emissions of sulfur are considerably lower in 2050 compared to 2000, which causes a reduction in the burden of anthropogenic sulfate (-65% in CNTRL) and the associated condensation sink of sulfuric acid in the Arctic atmosphere, which can be expected to facilitate the formation of new particles (Wyslouzil et al., 1991). The condensation sink of sulfuric acid is further reduced by increased wet deposition of aerosols due to increased Arctic precipitation. Finally, increased evaporation of moisture from the ocean leads to increases in relative humidity in the Arctic, which also produces conditions that are more favorable to nucleation in 2050 than 2000. On the other hand, increases in the sulfuric acid condensation sink due to increased emissions of sea salt and organic aerosols from the open ocean are not accounted for in the current version of the model, which may lead to overestimates in nucleation rates in the simulations. According to Browse et al. (2014), the increase in the natural condensation sink due to increased production of sea salt and organic aerosol under ice-free conditions causes a substantial reduction in the near-surface nucleation rate. However, it is likely that increases in production of sea salt under ice-free conditions are accompanied by large increases in wet deposition of sea salt due to increased Arctic precipitation (Struthers et al., 2011), which has not been explicitly accounted for by Browse et al. (2014). In addition, recent observations indicate a dominant role of small particles in activation and formation of cloud condensation nuclei in clean Arctic conditions (Leaitch et al., 2016) implying efficient nucleation of fine mode particles.

According to the first indirect effect of aerosols on climate, increases in CDNC may lead to smaller cloud droplets which are associated with more efficient scattering of incoming solar radiation and therefore stronger cloud radiative forcings, determined here as the difference in total-sky minus clear-sky shortwave radiative fluxes at top of the atmosphere (Soden et al., 2004). Subsequently, the cloud radiative forcing associated with biogenic DMS from Arctic DMS emissions is determined as difference in cloud radiative forcing between sensitivity experiments and CNTRL. As shown in Fig. 7 the cloud radiative forcing due to Arctic DMS emissions is small and not statistically significant for CLIM for both time periods. However, for 10×CLIM and 10×SAT the cloud radiative forcing in the Arctic due to Arctic DMS emissions is significant with maximum of up to -4 Wm$^{-2}$ for the Atlantic side of the Arctic for 10xCLIM in 2050, qualitatively in agreement with differences in CDNC. Overall, the mean cloud radiative forcing in the Arctic due to Arctic DMS emissions increases by between 108% (CLIM) and 145% (UNFM) from 2000 to 2050 (Table 2). All DMS data sets produce similar patterns of changes, with systematically enhanced cloud radiative forcings for the Atlantic region of the Arctic where loss of sea ice leads to particularly large increases in DMS emissions in all cases.

On regional scales, differences in cloud radiative forcing due to Arctic DMS emissions in Fig. 7 are generally smaller than changes in cloud radiative forcing associated with changes in meteorological conditions and anthropogenic aerosol precursor emissions between 2000 and 2050 (Fig. S4). However, averaged over the Arctic, differences are similar. For instance, the mean cloud radiative forcing in the Arctic in CLIM is -0.13 and -0.27 Wm$^{-2}$ for 2000 and 2050 respectively (difference of -0.14 Wm$^{-2}$). Similarly, CNTRL produces a difference in cloud radiative forcings of -0.65 Wm$^{-2}$ in total cloud radiative forcing between 2000 and 2050. It is evident that the cloud radiative forcing from Arctic DMS (Fig. 7) acts to enhance negative cloud radiative forcings in the central Arctic and counteracts positive forcings in the Atlantic Arctic and north of Siberia (Fig. S4), especially for 10xCLIM (the Arctic mean difference for 10×CLIM is -0.84 Wm$^{-2}$ between 2000 and 2050).

Mean results in the Arctic are summarized in Fig. 8 which provides an indication of robust aerosol and cloud responses at pan-Arctic scale, despite large differences in amount and spatial distribution of surface seawater DMS concentration in the different cases. For instance, Arctic-mean sulfate burdens due to Arctic DMS emissions are similar for present-day and future conditions (Fig. 8b) despite strongly increased DMS emissions in 2050 resulting from sea ice retreat. On the other hand, biogenic DMS emissions lead to more efficient formation of cloud droplets in the future in the Arctic. Therefore, cloud droplet number concentrations and cloud radiative forcing increase systematically as sea ice extent declines from 2000 to 2050 for each Arctic DMS, despite the low sensitivity of biogenic sulfate burdens to changes in sea ice. This provides evidence for a negative feedback of Arctic DMS emissions on Arctic radiative forcing, assuming that DMS concentrations in the ocean and atmospheric oxidant concentrations do not change between 2000 and 2050. To a good first approximation, the strength of the feedback is proportional to the mean surface seawater DMS concentration in the Arctic despite low sensitivity of sulfate burdens. The simulated responses of clouds and radiative forcing to changes in sea ice extent are found to be robust for a wide range of surface seawater DMS concentration scenarios.

## 5 Conclusions

Simulated sulfate concentrations from the Canadian Center for Climate Modeling and Analysis Atmospheric Model (CanAM4.3) were compared to observations from various shipboard campaigns and in-situ observations at Alert in Canada with a particular emphasis on the role of biogenic emissions. We found that the model reproduced seasonal variations in observed biogenic sulfur concentrations at Alert, although the model overestimates the biogenic contribution to total sulfate somewhat. Observed biogenic sulfur concentration maxima in May and in October are well reproduced by the model. Furthermore, comparisons with ship-based measurements from different field campaigns yield good agreement with simulated sulfate concentrations, especially in regions with large contributions of biogenic sulfate. However, it is plausible that the current Arctic surface seawater DMS concentrations are underestimated because models and climatological data sets do not yet account for substantially enhanced concentrations in melt ponds, and near the ice edge (e.g. Mungall et al., 2016; Ghahremaninezhad et al., 2016; Hayashida et al., 2017; Gourdal et al. 2018). In addition, large uncertainties exist for nucleation parameterizations (e.g. Zhang et al., 2010).

We performed model simulations to understand the sensitivity of sulfate aerosols and cloud radiative forcing to projected changes in sea ice and climate conditions between 2000 and 2050. Several model experiments were performed using a wide range of different surface seawater DMS concentrations in order to account for uncertainties in present-day and future DMS and to explore the sensitivity of aerosol/climate interactions to differences in spatial patterns of DMS. Results of the simulations indicate that the enhanced wet removal efficiency from increased precipitation in 2050 largely counteracts the impact of the increase in DMS emissions on sulfate burden in the Arctic. Annual mean Arctic sulfate burden differences between 2000 and 2050 are small for any given scenario (differences ranging between -1 and 21%) despite large increases in DMS emission between 2000 and 2050 due to sea ice retreat (between +33 and +53%). The sensitivity of modeled DMS fluxes into the atmosphere and sulfate burdens to spatial variations in surface seawater DMS is relatively weak.


Similar to previous studies (e.g. Browse et al., 2014; Ridely et al., 2016) we found weak impacts of climatological DMS emissions on cloud radiative forcings for present day conditions (simulation CLIM). However, for 2050 simulations, biogenic DMS emissions lead to considerable impacts on simulated Arctic aerosol and cloud processes owing to conditions that are conducive to the formation of fine particles in the Arctic. In 2050, increased emissions of DMS from

large ice-free regions of the Arctic ocean are associated with increased biogenic sulfate aerosol nucleation rates (between +128 and +269%) and cloud droplet number concentrations (between +35 and +133%) and thus enhanced cloud albedos, resulting in negative cloud radiative forcing of biogenic sulfate in the Arctic. The difference in cloud radiative forcing between years 2050 and 2000 based on simulations for four different Arctic surface seawater DMS data sets ranges from between -0.14 $Wm^{-2}$ (CLIM) to -0.84 $Wm^{-2}$ (10×CLIM). Thus our model results provide evidence for a negative Arctic

climate feedback. The essential ingredient of the feedback is a response of DMS emissions and cloud droplet number concentrations to sea ice retreat due to changes in radiative forcings in the climate system. This differs from CLAW which is rooted in the assumption of a change in biological production of DMS in the ocean in response to a change in radiative forcings. Furthermore, the strength of the Arctic climate feedback is proportional to the mean surface seawater DMS concentration in the Arctic. Consequently, potential future changes in primary productivity (Yool et al., 2005;

Vancoppenolle et al., 2013), mixing and phytoplankton habitat (Harada, 2016) in the Arctic Ocean (Levasseur, 2013) may act to enhance the strength of the Arctic feedback.

The model simulations used in the current study are not interactively coupled with ocean and sea ice DMS and therefore rely on specified surface seawater DMS concentrations. The current model version does not account for increases in

sulfuric acid condensation sink due to increased emissions of sea salt and organic aerosols from the open ocean, which may lead to overestimates in nucleation rates in the simulations. However, there is no consensus on the CCN activity of sea spray aerosols (including primary organic aerosols and sea salts; Neukermans et al., 2018). Based on historical shipboard observations, Quinn et al. (2017) concluded that a small fraction of marine cloud condensation nuclei are made up of sea spray aerosol especially in regions north of 60N. Leaitch et al. (2016), based on recent observations in

the Arctic region, also found that small particles (up to 20nm) are activated in summer. Similarly, Collins et al. (2017) reported frequent occurrence of activation of ultrafine particles in the Canadian Arctic Archipelago. Additional uncertainty in the strength of the feedback arises from the fact that atmospheric oxidant concentrations are assumed to be steady in our study. More comprehensive assessments of the strength and impacts of DMS/climate feedbacks in the Arctic will become possible once a new generation of Earth System Models with interactive ocean and sea ice DMS,

chemistry, and climate processes becomes available.

**Author contribution**

K. v. S. and R. M. conceived and designed the study and wrote the paper. R. M. undertook the analyses and produced the

figures. A.-L. N. provided the isotopic data and M. G. provided a summary of oceanic DMS processes. All co-authors helped edit the paper.

**Acknowledgement**

Shipping non-sea-salt sulfate data was obtained from NOAA PMEL Atmospheric Data Server (available at: *https://saga.pmel.noaa.gov/data/*). We thank Jim Christian and two anonymous referees for providing helpful reviews of the manuscript. We also thank Martine Lizotte and Hakase Hayashida for providing very helpful discussions of surface ocean DMS concentrations. Funding for this work was provided by the Natural Sciences and Engineering Research Council of Canada through the NETCARE project of the Climate Change and Atmospheric Research Program.

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

Table 1. Arctic mean surface seawater DMS concentration, total sulfur emission flux, and associated mean biogenic sulfate burden for 2000 (2050)

|  | DMS (nM) | Emissions (TgS/yr) | $SO_4^{2-}$ Burden (kilotonnes) |
|---|---|---|---|
| CLIM | 1.96 | 0.24 (0.32) | 2.13 (2.58) |
| 10×CLIM | 19.58 | 2.41 (3.22) | 20.05 (20.55) |
| UNFM | 16.88 | 1.88 (2.87) | 16.82 (16.64) |
| 10×SAT | 11.19 | 1.31 (1.92) | 10.31 (11.63) |

Table 2. Arctic mean aerosol nucleation rate, CDNC, and cloud radiative forcing in 2000 (2050) associated with emissions of DMS in the Arctic.

|  | Nucleation rate ($\times 10^6$ m$^{-2}$s$^{-1}$) | CDNC ($\times 10^6$ m$^{-3}$) | Cloud radiative forcing (Wm$^{-2}$) |
|---|---|---|---|
| CLIM | 0.02517 (0.09294) | 0.34716 (0.81023) | -0.13 (-0.27) |
| 10×CLIM | 0.41751 (0.98998) | 2.9886 (4.1781) | -0.75 (-1.59) |
| UNFM | 0.27358 (0.62366) | 2.3019 (3.4893) | -0.40 (-0.98) |

| | 10×SAT | 0.20618 (0.54223) | 1.7853 (2.4045) | -0.55 (-1.18) |
|---|---|---|---|---|

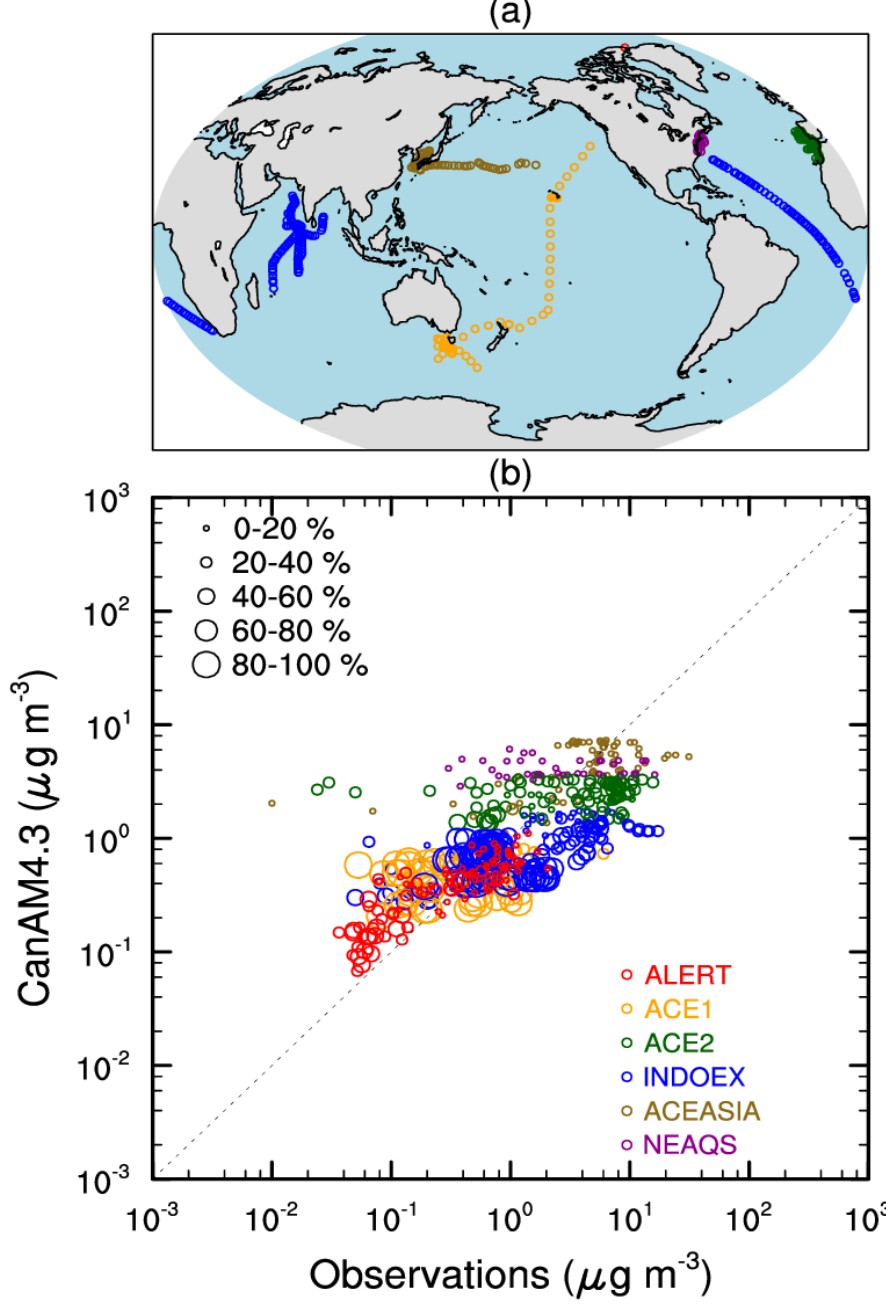

**Figure 1: (a) Locations for Alert (red circle) and ship-based nss-SO$_4^{2-}$ observations used for comparisons with CanAM4.3? model results. (b) Comparison of model and observed nss-SO$_4^{2-}$. For ship based observations, the size of the markers represent percentage of contribution of DMS to total nss-SO$_4^{2-}$ derived from model results. For Alert, the percentage contribution to total nss-SO$_4^{2-}$ is based on isotopic composition.**

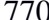

**Figure 2: Multi-year mean monthly concentrations of total (a), fossil-fuel (b), biogenic (c) sulfate concentrations from simulations and observed nss-SO$_4^{2-}$ during 1994 to 2002 at Alert, (d) relative contribution of DMS source to total sulfate concentration at Alert. The whiskers represent minimum and maximum and the horizontal line inside the box represents the mean for the whole period. The box height represents the interquartile range of 25$^{th}$ and 75$^{th}$ percentiles. Unit: ng m$^{-3}$**

**Figure 3: Annual mean distribution of surface seawater DMS concentration used in sensitivity simulations; (a) L10 DMS climatology, (b) L10 DMS climatology multiplied by 10 in the Arctic region, (c) uniform distribution of DMS, (d) satellite based DMS climatology multiplied by 10. The bottom panel shows zonal mean results.**

800

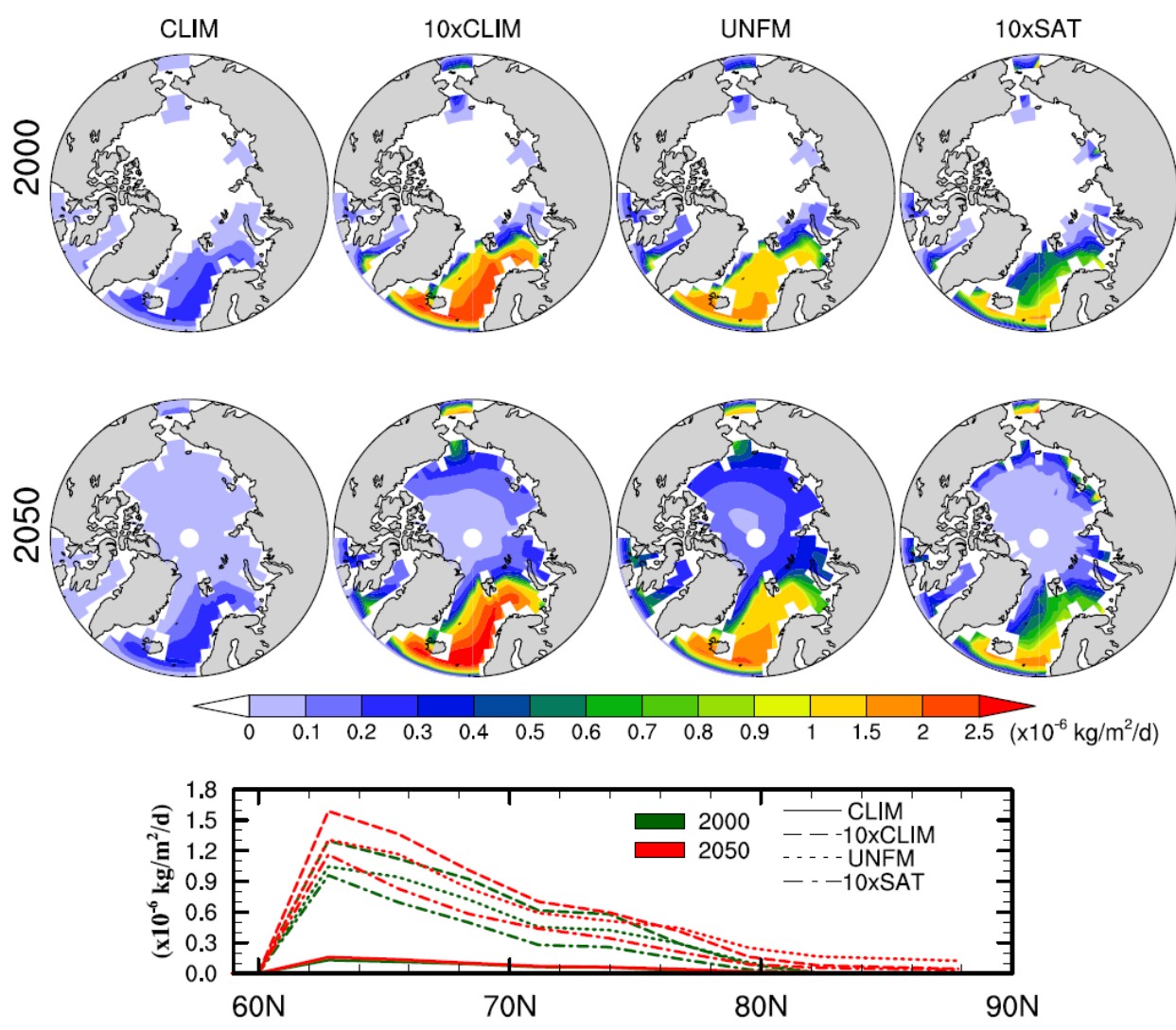

**Figure 4: DMS emission fluxes for the two different periods, similar to Fig. 3. Zonal mean results 2000 and 2050 are shown in the bottom panel.**

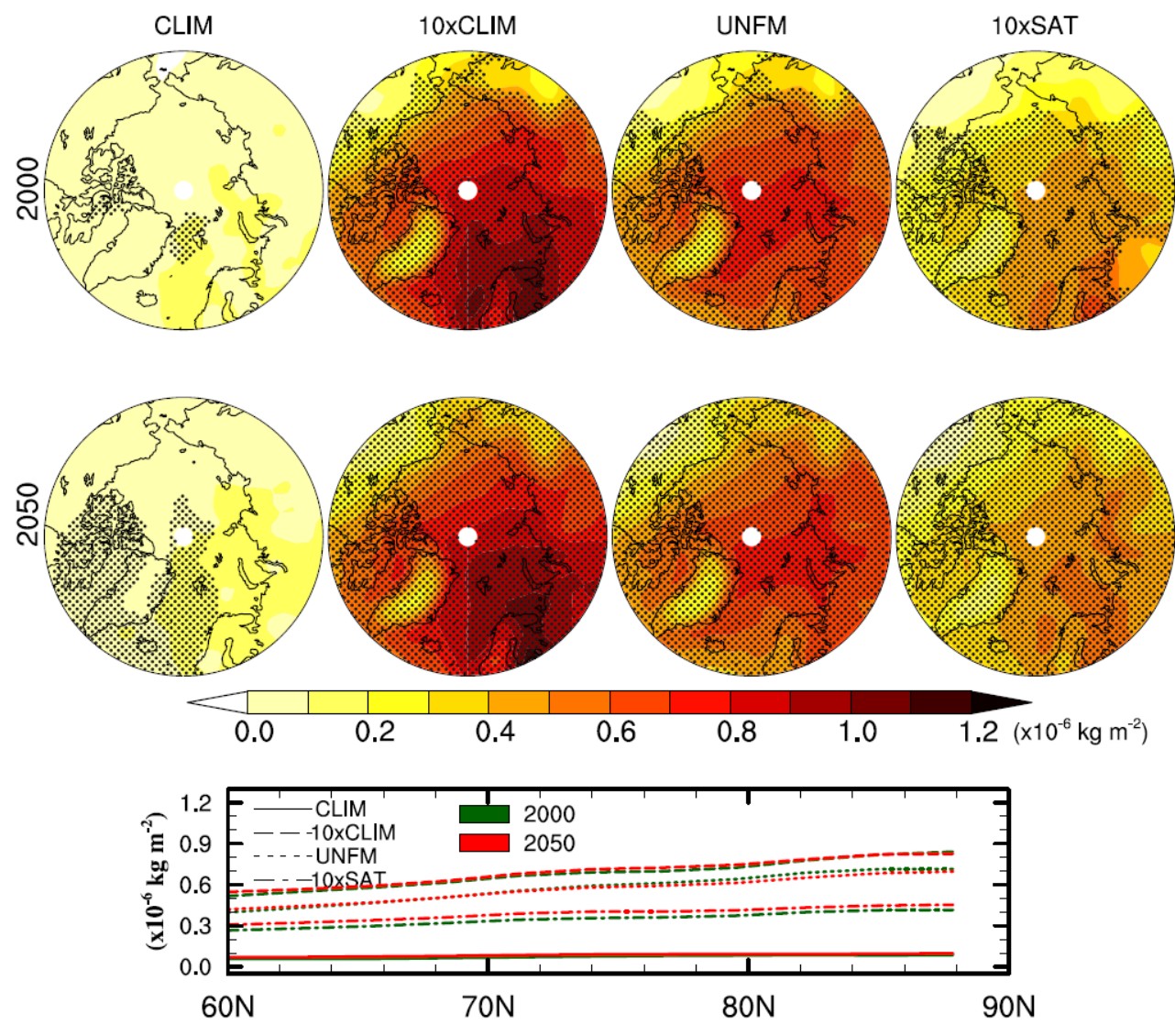

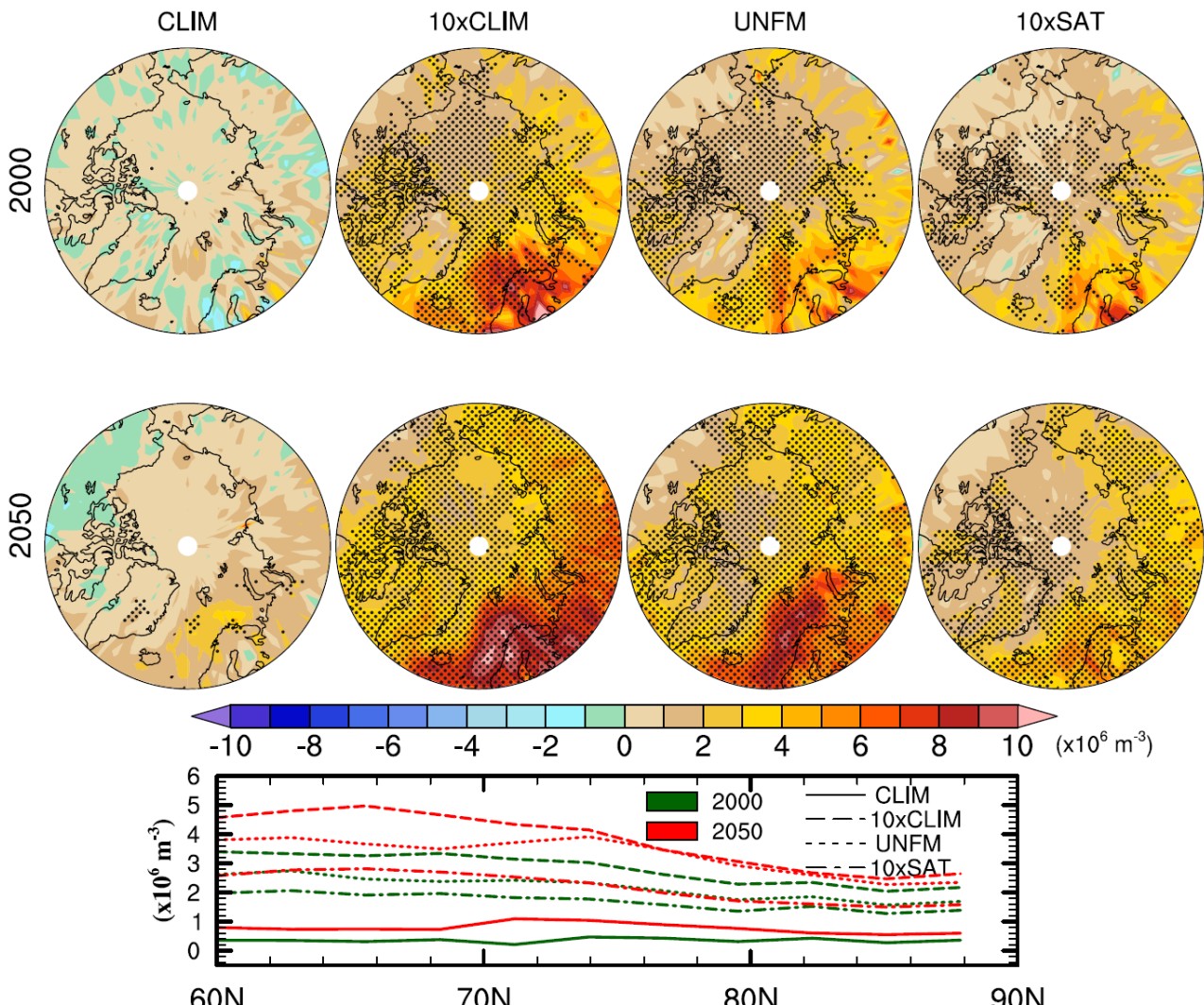

**Figure 5: Biogenic sulfate burden due to Arctic DMS emissions for each of the 4 scenarios. The stippling represents regions where the burden difference relative to CNTRL is significant at 95% confidence level. Zonal mean results for 2000 and 2050 are shown in the bottom panel.**

**Figure 6: Change in cloud droplet number concentration at first model level above the surface due to Arctic DMS emissions (relative to CNTRL). Stippling represents change significant at 95% confidence level. Zonal mean results are shown in the bottom panel.**

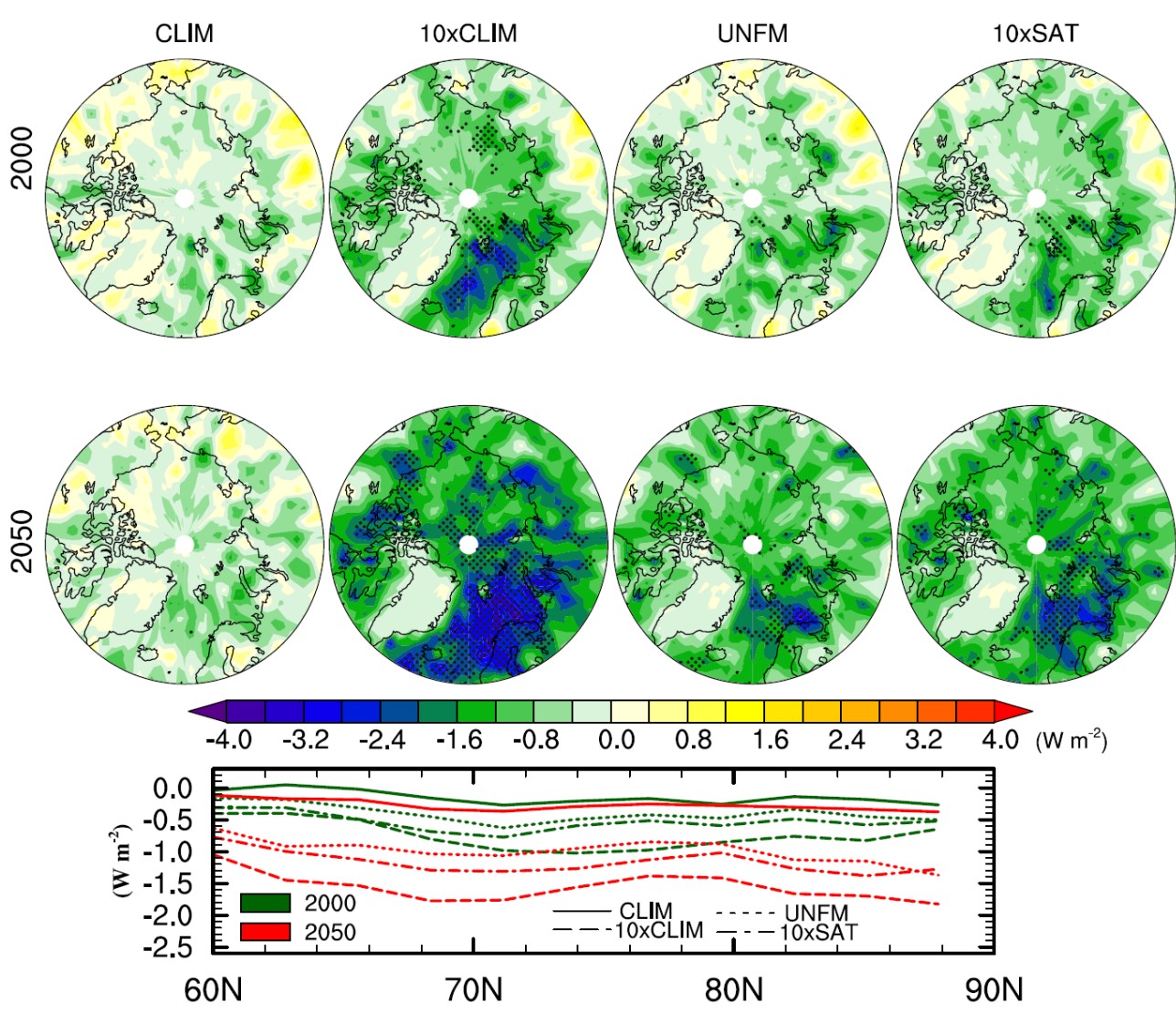

Figure 7: Cloud radiative forcing in the Arctic due to ocean DMS emissions. Stippling represents radiative forcing significant at 95% confidence level. Zonal mean results are shown in the bottom panel.

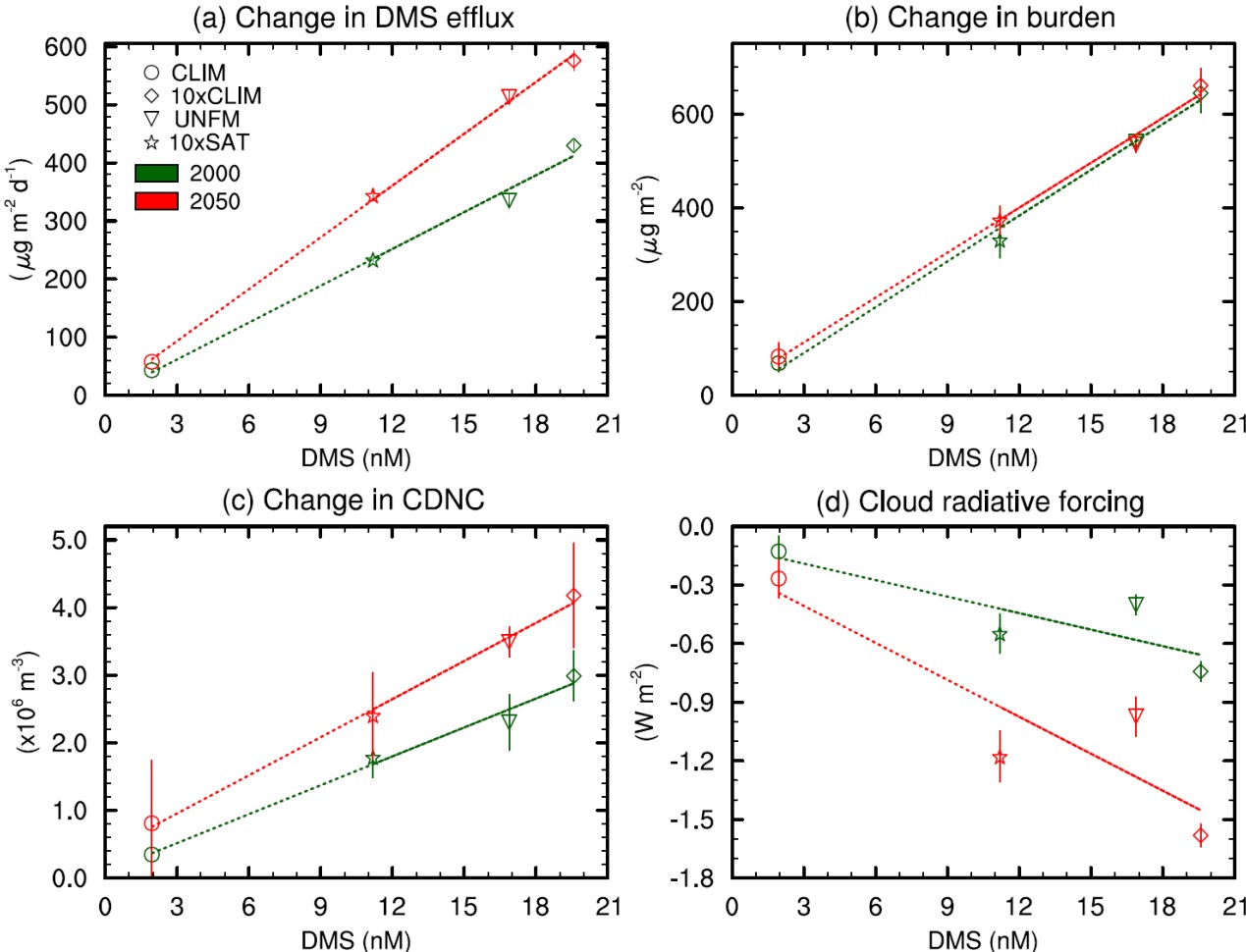

**Figure 8: Relationship between annual Arctic mean DMS emission fluxes, sulfate burden, CDNC, and cloud forcing and the mean DMS concentration in the Arctic. The vertical lines represent 95% confidence interval based on two-tailed t-test. Dotted lines represent regression between four scenarios.**