# Peer review of "Sensitivity of Arctic sulfate aerosol and clouds to changes in future surface seawater dimethylsulfide concentrations"

_Atmospheric Chemistry and Physics, 2018_

## Referee Comment (RC1) · Anonymous Referee #1 · 12 Nov 2018

This study concerns the impact of Arctic DMS emission on sulfate aerosol concentrations and cloud formation under present and future sea-ice conditions. The authors use a model nudged to winds and temperature for year 2000 and 2050 with various assumptions for DMS concentrations. DMS emissions are highly uncertain and modeling studies of the impact of DMS emissions are still few, and I think this study makes a useful contribution to the field. The perturbations of DMS concentrations are quite high, but the authors argue well that such high perturbations are needed. The conclusion about the negative feedback loop along with figure 8 are interesting findings. I would recommend this manuscript for publication after some clarifications given below sorted by line number:

L40: Could you give examples of such favorable conditions in the atmosphere and ocean?

L51: Could you add why there is little evidence under present climate conditions?

L52: Which important feedback loops exists in the Arctic?

L57: Again, why would the emissions be enhanced particularly in the Arctic?

L63: How is this shown to be problematic?

L71: How do the Arctic DMS concentrations from Lara et al. 2011 differ from previous studies?

L111: I am a little confused about these sensitivity tests: are they linked to your study or are they different? If the latter; what did they show and how is your study different from them?

L158: Testdal/Tesdal reference

L160: Would the nudging to temperature and winds influence how the perturbed DMS emissions impact clouds compared to running with free meteorology?

L185: 'good agreement': a bit vague; could you add a number here?

L231: Why did you choose this number (16.9nM)?

L247: Can you also add that you are using the sulfur emissions from RCP8.5 as well for year 2050? (if that is what you are using?)

Figure 4-8: it is a bit difficult to separate the lines for the different runs in the panel which shows the zonal means. Also; why do you show 60N-90N for fig 4, and NH for the others? As far as I can see, you don't discuss the results south of the Arctic, so I suggest only showing the Arctic latitudes -and make the plots larger, the lines thicker and/or different colors to make it easier to distinguish the lines.

L274: Can you add the numbers for increased precipitation and wet removal?

L281: How large are the reductions?

L290: Do you think you would get a significant signal if the run was longer?

L307: Could you remind us of what the condensation sink is in the model?

L382: ‘However, in the future’: change to ‘for year 2050 simulations’ or similar?

L399: Can you add a couple of sentences about the main uncertainties in your model + set-up and link this to the last sentence?

---

## Referee Comment (RC2) · Anonymous Referee #2 · 9 Feb 2019

The presented study uses an atmospheric global climate model to determine the impacts of increased oceanic DMS emissions on the Arctic sulfate aerosol budget, cloud droplet number concentration (CDNC) and cloud radiative forcing. The main finding is that increased wet scavenging of sulfate in 2050 compensates for any increased aerosol production due to higher DMS emissions. Furthermore, significantly higher CDNC and more negative cloud radiative forcing are found in 2050 because of higher nucleation rates. The first half of the paper describes the comparison of a historical simulation (1991-2003) with observations from ship-based campaigns and at Alert in the Canadian Arctic. The second half of the paper reports results from a simulation experiment for 2000 and 2050. The illustrations and visualization of results adequately

reflect the key findings of the analysis. The first half is rather well presented and written in a comprehendible way. The second half (Sect. 4) is very difficult to evaluate because of a lack of information and a confusing description of the simulations. The manuscript should have a section (for example after Sect. 2) that provides an overview of all details and motivation behind the simulations in one place. Moreover, it is unclear why the historical simulation was not used as present-day reference simulation for the future runs, but instead a new one, "2000", integrated over four years, is introduced on page 7. How can we be sure that "2000" has the same robustness as the historical simulation?

Scientifically, the study is questionable due to three major issues:

1) The future state of the atmosphere in 2050 is derived from integrated simulations of four years, 2048-2051, which seems to be too short. It appears that the climate simulation was meant to follow the RCP8.5 emission scenario. Have atmospheric concentrations of $CO_2$ and other trace gases as well as the global air temperatures in 2050 reached levels that are comparable to multi-decadal climate simulations for RCP8.5? In case this has been achieved during the 4-year run: has the output from the spin-up period been excluded from the analysis?

2) Since the increase of CDNC due to increased emissions of DMS in 2050 compared to 2000 was significant, I would expect that the cloud microphysics are likewise significantly influenced by the higher number of cloud condensation nuclei (CCN), as this has been demonstrated for the summertime Arctic (Leaitch et al., 2013). If a cloud forms on a higher number of CCN the condensed water will be distributed over many small droplets rather than over a few large ones, given that the available amount of water is the same. An increase in CCN concentration results in faster evaporation rates owing to smaller cloud droplets. The faster evaporation rate leads to enhanced entrainment of sub-saturated air surrounding the cloud and a decrease in cloud fraction, in turn lowering the aerosol effect on cloud albedo (Zuidema et al., 2008). Have such aerosol-induced changes on the cloud macrophysics be considered?

3) Sea-salt particles and primary organic particles could be much more efficient CCN than particles derived from DMS (Quinn and Bates, 2011). If the sea salt or organic particle was already sufficiently large to serve as a CCN, the addition of DMS-derived sulfur to the particle will not increase the number of CCN. The increase of the primary sea-spray emissions with retreating ice, would not just be compensated by increased wet scavenging, but might outcompete DMS as precursor for CCNs. Clearly, increased emission of sea-salt aerosol will inhibit the development of precipitation. It will also cause more large CCNs, which can efficiently suppress activation of some of the smaller (sulfate) particles (O'Dowd et al. 1999).

Specific Points:

- P.1 line 41: Mention that gaseous MSA also nucleates and plays an important role in the initial growth of new particles. Importantly, a recent study in the Canadian Arctic Archipelago by Willis et al. (2016) presents observational evidence that the growth of nucleation mode aerosol in the summertime Arctic is correlated with the presence of particulate MSA and organic species.

- P.2 line 71: DMS in water or in air. Suggest to denote seawater DMS as DMS(aq) and gaseous DMS as DMS(g).

- P.3 line 98: How much of the produced DMSP is transferred to sediments? Does is not depend on grazing pressure how much DMS is actually produced? How sensitive are diatoms and haptophytes to seawater temperature changes?

- P.3 line 108: Please explain why a factor of 10 is used. Are there any projections about future DMS emission (from the same water column, not due to ice loss) that justify this order of magnitude increase?

- P.4 line 127: How well does Piecewise Lognormal Approximation cope with newly introduced particles from nucleation?

- P.4 line 130: In high latitude regions - characterized by low temperatures and high

wind speeds - estimated DMS transfer velocity from wind speed parameterizations will be biased high if only the Schmidt number normalization is used.

- P.4 line 131: Does the model include oceanic emissions of sea-salt particles and primary organic particles? If not, that must be stated here.

- P.4 line 134 - 135: "MSA is treated as sulfuric acid in model for simplicity" - this is problematic in multiple ways. MSA forms in one step from the oxidation of DMS. Given it nucleates as sulfuric acid, then the atmospheric nucleation would be much too efficient. Please provide the average nucleation rates of Table 2 with literature data from the Arctic, e.g. Karl et al. (2012) and Leaitch et al. (2013). Although MSA is much less efficient than sulfuric acid in forming particles with water molecules, several laboratory studies and computational studies confirmed that MSA forms particles with alkylamines (Dawson et al., 2012; Chen et al., 2015; Xu et al., 2018). The presence of water seems to control the new particle formation in this system.

- P. 6, line 190 - 194: What explains the high modelled biogenic sulfate in June in Fig. 2c? Can it be related to the DMS seawater concentrations?

- P. 6, line 205: Please explain the occurrence of the October peak in the annual cycle.

- P. 6, line 217: Please provide more details about the radiative flux calculations in Sect. 2. Which parameters were perturbed? A table with parameters and perturbation values for all 25 simulations would be very helpful.

- P. 7, line 241: Explain better what "corresponding simulation" means here.

- P. 7, line 242-244: An explanation is missing here, why a new simulation "2000" and not the historical simulation "hisCont" was used as present-day reference.

- P. 8 line 286: There could be two reasons why wet deposition increased, growth of particles to larger sizes by condensation of DMS oxidation products which makes them more accessible to wet scavenging or the increase of precipitation rates. Please provide information on the average precipitation rates in the simulations in Table 2.

Also give the average liquid water content of clouds in Table 2.

- P. 8 line 295: Is "10ˆ7 mˆ-3" the number change or the absolute number of CDNC? Please set the value in relation to average CDNC in the present-day simulation.

- P. 11 line 375: How much does in-cloud sulfate production increase? More details on aqueous-phase production of sulfate in the model and about treatment of in-cloud scavenging should be provided in Sect. 2.

- P. 11 line 390 - 395: Could you elaborate on the expected feedbacks due to increased SST? Warmer water would be less favorable for diatoms but the solubility of DMS would be lower.

- P. 11 line 396-397: Steady-state atmospheric oxidant concentrations are not the only additional uncertainty. The assumptions on nucleation rates and in-cloud scavenging of sulfate seem to be critical for the conclusion of this study.

References:

Chen, H., Ezell, M. J., Arquero, K. D., Varner, M. E., Dawson, M. L., Gerber, R. B., and Finlayson-Pitts, B.: New particle formation and growth from methanesulfonic acid, trimethylamine and water, Phys. Chem. Chem. Phys., 17, 13699, doi:10.1039/c5cp00838g, 2015.

Dawson, M. L., Varner, M. E., Perraud, V., Ezell, M. J., Gerber, R. B., and Finlayson-Pitts, B.: Simplified mechanism for new particle formation from methanesulfonic acid, amines, and water via experiments and ab initio calculations, Proc. Natl. Acad. Sci. USA, 109(46), 18719-18724, doi:10.1073/pnas.1211878109, 2012.

Karl, M., Leck, C., Gross, A., and Pirjola, L.: A study of new particle formation in the marine boundary layer over the central Arctic Ocean using a flexible multicomponent aerosol dynamic model, Tellus B, 64, 17158, doi:10.3402/tellusb.v64i0.17158, 2012.

Leaitch, W. R., Sharma, S., Huang, L., Toom-Sauntry, D., Chivulescu, A., Macdonald, A. M., von Salzen, K., Pierce, J. R., Bertram, A. K., Schroder, J. C., Shantz, N. C., Change, R. Y.-W., and Norman, A.-L.: Dimethyl sulfide control of the clean summertime Arctic aerosol and cloud, Elementa: Science of the Anthropocene, 1(17), doi:10.12952/journal.elementa.000017, 2013.

O'Dowd, C. D., Lowe, J. A., Smith, M. H., and Kaye, A. D.: The relative importance of non-sea sulphate and sea-salt aerosol to the marine cloud condensation nuclei population: an improved multi-component aerosol-cloud droplet parameterization, Q. J. R. Meteorol. Soc., 125, 1295–1313, 1999.

Willis, M. D., Burkart, J., Thomas, J. L., Köllner, F., Schneider, J., Bozem, H., Hoor, P. M., Aliabadi, A. A., Schulz, H., Herber, A. B., Leaitch, W. R., and Abbatt, J. P. D.: Growth of nucleation mode particles in the summertime Arctic: a case study, Atmos. Chem. Phys., 16, 7663–7679, doi:10.5194/acp-16-7663-2016, 2016.

Xu, J., Perraud, V., Finlayson-Pitts, and Gerber, R. B.: Uptake of water by an acid-base nanoparticle: theoretical and experimental studies of the methanesulfonic acid-methylamine system, Phys. Chem. Chem. Phys., 20, 22249, doi:10.1039/c8cp03634a, 2018.

Zuidema, P., Xue, H., and Feingold, G.: Shortwave radiative impacts from aerosol effects on marine shallow cumuli, J. Atmos. Sci., 65, 1979–1990, 2008.

---

## Author Comment (AC1) · 1 Apr 2019

This study concerns the impact of Arctic DMS emission on sulfate aerosol concentrations and cloud formation under present and future sea-ice conditions. The authors use a model nudged to winds and temperature for year 2000 and 2050 with various assumptions for DMS concentrations. DMS missions are highly uncertain and modeling studies of the impact of DMS emissions are still few, and I think this study makes a useful contribution to the field. The perturbations of DMS concentrations are quite high, but the authors argue well that such high perturbations are needed. The conclusion about the negative feedback loop along with figure 8 are interesting findings. I would recommend this manuscript for publication after some clarifications given below sorted by line number:

Response: We are very thankful to the reviewer for reviewing our paper and providing very useful comments that lead to the overall improvement of the manuscript. Replies to the referee's comments are given below following the individual comment.

L40: Could you give examples of such favorable conditions in the atmosphere and ocean?

Response: This includes all processes enhancing turbulence at the air-water interface: wave break (in turn depending on wave types), bubbles, surface films, friction velocity, humidity and temperature gradients, convection, and ice shear (Jahne et al., 1987; Merlivat and Memery, 1983; Memery and Merlivat, 1985; Monahan and Spillane, 1984; Wanninkhof et al., 2009; Loose et al., 2014; van der Loeff et al., 2014). Please see our edits in the manuscript.

Although gas exchange is generally parameterized as a function of wind speed for convenience, this approach is clearly an oversimplifcation.

L51: Could you add why there is little evidence under present climate conditions?

Response: A review of several years of observations, laboratory experiments and modelling studies was reported by Quinn and Bates (2011) who argued that the sources of cloud condensation nuclei in marine boundary layer are numerous and the response of clouds to changes in aerosols are more complex than previously thought at the time of CLAW hypothesis. This is now reflected in the manuscript.

L52: Which important feedback loops exists in the Arctic?

Response: As found by Leaitch et al. (2013), DMS is an important source of new particle formation during cleaner Arctic summer. These newly formed particles influence cloud albedo through cloud microphysical processes and thus can influence local radiation budgets and feedback mechanisms.

L57: Again, why would the emissions be enhanced particularly in the Arctic?

Response: The loss of Arctic sea-ice allows further penetration of sunlight into surface ocean water (e.g. Nicolaus et al., 2012) that can increase net production of algae and phytoplankton (e.g. Arrigo and van Dijken, 2015). In addition, when sea-ice is melted the surface ocean water is more prone to wind stress that can enhance air-fluxes of trace gases (e.g. Bates et al., 2006; Ardyna et al., 2014). However, as discussed in the introduction section of the paper, future changes in net primary production and surface ocean water DMS concentrations are uncertain. Please see our changes in the manuscript.

L63: How is this shown to be problematic?

Response: This has been elaborated in the subsequent lines which read as follows:

*"The small number of measurements from other locations in the Arctic is problematic, as recent research in the NETCARE network (Abbatt et al., 2018) has shown. Surface seawater DMS concentrations measured in the Canadian Arctic in July and August of 2014 and 2016 were substantially higher than those used by Lana et al. for July and August (e.g., NETCARE median concentrations of 4.4 nmol/L and 7.3 nmol/L, Martine Lizotte, personal communication; median concentration range from 0.5 to 4.4 nmol/L for Lana et al., https://saga.pmel.noaa.gov/dms/). Furthermore, melt ponds on sea ice represent a yet missing source of DMS in studies of the Arctic (Mungall et al., 2016; Ghahremaninezhad et al., 2016; Gourdal et al. 2018; Abbatt et al., 2018). Gali et al. (2018) argue that biases in the climatology by Lana et al. arise from the application of objective interpolation procedures to a limited amount of measurements. Consequently, Arctic DMS concentrations based on Lana et al. (2011) differ substantially from those of an earlier climatology (Kettle and Andreae, 2000), ocean biogeochemical models, and DMS parameterizations (Tesdal et al., 2016a), indicating large uncertainties in estimates of surface seawater DMS concentrations."*

L71: How do the Arctic DMS concentrations from Lara et al. 2011 differ from previous studies?

Response: Annual mean concentration differences between Kettle and Andreae (2000) and Lana et al. (2011) include zonal mean higher concentrations in Kettle and Andreae (2000) in polar regions of both hemispheres. For the Arctic region (>60N) the emission fluxes are 25% less in Lana et al. (2011) using parameterization of Nightingale et al. (2000).

L111: I am a little confused about these sensitivity tests: are they linked to your study or are they different? If the latter; what did they show and how is your study different from them?

Response: Our study is completely independent of previous studies in terms of model used for experiment, the study region, and different perturbation scenarios/experiments. While our study focusses on the Arctic region, Fiddes et al. (2018) focussed on southern hemisphere and Grandey and Wang (2015) performed experiments relevant to artificial cooling effects as a potential geo-engineering solution to global warming. In our study we analyzed the sensitivity of sulfate aerosol concentrations and changes in cloud microphysical properties using different scenarios of ocean DMS concentrations. As mentioned in the introduction, we used a state-of-the-art climate model and the experiments are described in detail in the following sections.

L158: Testdal/Tesdal reference
Response: Thank you, this is corrected.

L160: Would the nudging to temperature and winds influence how the perturbed DMS emissions impact clouds compared to running with free meteorology?

Response: No, this is not the case. We previously investigated the impact of different nudging strategies on simulated radiative forcings and compared with results from simulations without nudging. Based on these tests, we are confident that the simulated radiative forcings are meaningful. Only emissions of DMS are perturbed, which affects sulfate aerosol concentrations but does not lead to significant changes in concentrations of other radiative forcing agents in the model. Changes in sulfate concentrations lead to changes in cloud albedo via the first indirect effect in the simulations. Atmospheric temperatures and winds in the simulations are already strongly constrained by the use of specified SSTs and sea ice fraction in the model, which allows us to study the impact of DMS emissions for specified climate conditions for present-day and in the future. We do not apply any nudging to the simulated specific humidity and cloud water content so cloud radiative forcings are not directly affected by nudging. Diagnosed radiative effects from our study provide a foundation and benchmark for future studies with fully coupled Earth System Models which can be used to simulate feedbacks in sea ice and global climate to changes in DMS emissions.

L185: 'good agreement': a bit vague; could you add a number here?

Response: This sentence has been removed. The numbers for model and observation comparisons are provided in the same paragraph and the two subsequent paragraphs.

L231: Why did you choose this number (16.9nM)?

Response: This number is based on grid-point average of 10*Lana etal. Climatology – please see Figure 3 for reference. The number is arbitrary and provides an additional scenario for testing the sensitivity of forcings under uniform surface seawater concentrations of DMS and also provides a test of linearity of such responses.

L247: Can you also add that you are using the sulfur emissions from RCP8.5 as well for year 2050? (if that is what you are using?)

Response: As described in lines 141-143, the aerosols and precursor emissions are from RCP4.5 for future simulations.

Figure 4-8: it is a bit difficult to separate the lines for the different runs in the panel which shows the zonal means. Also; why do you show 60N-90N for fig 4, and NH for the others? As far as I can see, you don't discuss the results south of the Arctic, so I suggest only showing the Arctic latitudes -and make the plots larger, the lines thicker and/or different colors to make it easier to distinguish the lines.

Response: We Agree and appreciate the referee for these suggestions. The figures have been revised accordingly.

L274: Can you add the numbers for increased precipitation and wet removal?

Response: This is included in Table S1 of the supplementary materials file.

L281: How large are the reductions?

Response: These are summarized in Table S1.

L290: Do you think you would get a significant signal if the run was longer?

Response: Given the large variability in CDNC in the simulations it seems unlikely that we would get a statistically significant difference in CDNC even if we extended the simulation by several decades. Although it would be possible to increase the significance of the results by extending the runs, this is not feasible from a practical point of view given the considerable costs of running the model. Also, this may not yield much new information since the differences in CDNC are likely to be small given that changes in DMS emissions are small. Furthermore, as we show in Fig. 8, we are able to obtain a robust relationship between cloud radiative forcing and DMS emissions despite relatively large variability in the results for near-surface CDNC.

L307: Could you remind us of what the condensation sink is in the model?

Response: Sulfuric acid gas condenses efficiently at the surface of aerosol particles owing to the low volatility of the gas. The rate of gas-to-particle transfer by condensation is determined by rates of diffusion and surface accommodation. If the rate of condensation of sulfuric acid is reduced, e.g. in an environment with low aerosol surface area, the formation of new aerosol particles is increased according to binary homogeneous nucleation theory (if nothing else is changed). We added a brief explanation in the text.

L382: 'However, in the future': change to 'for year 2050 simulations' or similar?

Response: This is changed as suggested, "for 2050 simulations". Thank you.

L399: Can you add a couple of sentences about the main uncertainties in your model
+ set-up and link this to the last sentence?

Response: Agree. The last paragraph of the manuscript is revised to incorporate model uncertainties, the paragraphs reads as:

"*The model used in the current study is not interactively coupled with ocean and sea ice DMS and therefore rely on specified surface seawater DMS concentrations. The increases in sulfuric acid condensation sink due to increased emissions of sea salt and organic aerosols from the open ocean are not accounted for in the current version of the model, which may lead to overestimates in nucleation rates in the simulations. Additional uncertainty in the strength of the feedback arises from the fact that atmospheric oxidant concentrations are assumed to be steady in our study. More comprehensive assessments of the strength and impacts of DMS/climate feedbacks in the Arctic will become possible once a new generation of Earth System Models with interactive ocean and sea ice DMS, chemistry, and climate processes becomes available.*"

References:

Ardyna, M., M. Babin, M. Gosselin, E. Devred, L. Rainville, and J.-É. Tremblay (2014), Recent Arctic Ocean sea ice loss triggers novel fall phytoplankton blooms, Geophys. Res. Lett., 41, doi:10.1002/2014GL061047.

Arrigo, K.R., van Dijken, G.L., 2015. Continued increases in Arctic Ocean primary production. Prog. Oceanogr. 136, 60–70.

Bates, N. R., Moran, S. B., Hansell, D. A., & Mathis, J. T. (2006). An increasing CO2 sink in the Arctic Ocean due to sea-ice loss. Geophysical Research Letters, 33(23).

Fiddes, S. L., Woodhouse, M. T., Nicholls, Z., Lane, T. P., and Schofield, R.: Cloud, precipitation and radiation responses to large perturbations in global dimethyl sulfide, Atmos. Chem. Phys., 18, 10177-10198, https://doi.org/10.5194/acp-18-10177-2018, 2018.

Grandey, B. S., and Wang, C.: Enhanced marine sulphur emissions offset global warming and impact rainfall, Sci. Rep., 5, 13055, doi:10.1038/srep13055, 2015.

Jahne, B., Munnich, K.O., Bosinger, R., Dutzi, A., Huber, W., and Libner, P.: On the parameters influencing air-water gas exchange, J . Geophys. Res., 92, 1937-1949, 1987.

Kettle, A. J., and Andreae, M. O.: Flux of dimethylsulfide from the oceans: A comparison of updated data sets and flux models, J. Geophys. Res.-Atmos., 105, 26793–26808, doi:10.1029/2000JD900252, 2000.

Lana, A., Bell, T. G., Simó, R., Vallina, S. M., Ballabrera-Poy, J., Kettle, A. J., Dachs, J., Bopp, L., Saltzman, E. S., Stefels, J., Johnson, J. E., and Liss, P. S.: An updated climatology of surface dimethlysulfide concentrations and emission fluxes in the global ocean, Global Biogeochem. Cycles, 25, GB1004, doi:10.1029/2010GB003850, 2011.

Leaitch, W. R., Sharma, S., Huang, L., Macdonald, A. M., Toom-Sauntry, D., Chivulescu, A., von Salzen, K., Pierce, J. R., Shantz, N. C., Bertram, A., Schroder, J., Norman, A.-L., and Chang R. Y.-W.: Dimethyl sulphide control of the clean summertime Arctic aerosol and cloud, Elementa: Science of the Anthropocene, 1, 000017, doi:10.12952/journal.elementa.000017, 2013.

Loose, B., McGillis, W. R., Perovich, D., Zappa, C. J., & Schlosser, P. (2014). A parameter model of gas exchange for the seasonal sea ice zone. Ocean Science, 10(1), 17-28.

Memery, L., and Merlivat L.: Modeling of gas flux through bubbles at the air-water interface, Tellus Ser. B, 37, 272-285, 1985.

Merlivat, L., and Memery L.: Gas exchange across an air-water interface: Experimental results and modeling of bubble contribution to transfer, J. Geophys. Res., 88, 707-724, 1983.

Monahan, E. C., and Spillane M. C.: The role of whitecaps in air-sea gas exchange, in Gas Transfer at Water Surfaces, edited by W. Brutsaert, and G.H. Jirka, pp. 495-504, D. Reidel, Norwell, Mass., 1984.

Nicolaus,M., Katlein, C.,Maslanik, J., Hendricks, S., 2012. Changes in Arctic sea ice result in increasing light transmittance and absorption. Geophys. Res. Lett. 39, L24501. http://dx.doi.org/10.1029/2012GL053738.

Nightingale, P. D., Malin, G., Law, C. S., Watson, A. J., Liss, P. S., Liddicoat, M. I., Boutin, J., and Upstill-Goddard, R. C.: In situ evaluation of air-sea gas exchange parameterizations using novel conservative and volatile tracers, Global Biogeochem. Cy., 14, 373y., 14doi:10.1029/1999GB900091, 2000.

Quinn, P. K., and Bates, T. S.: The case against climate regulation via oceanic phytoplankton sulphur emissions, Nature, 480, 51–56, doi:10.1038/nature10580, 2011.

Rainville, L., Lee, C. M., and Woodgate, R. A.: Impact of wind-driven mixing in the Arctic Ocean, Oceanography, 24(3), 136-145, 2011.

van der Loeff, M. M. R., Cassar, N., Nicolaus, M., Rabe, B., & Stimac, I. (2014). The influence of sea ice cover on air-sea gas exchange estimated with radon-222 profiles. Journal of Geophysical Research: Oceans, 119(5), 2735-2751.

Wanninkhof, R., Asher, W. E., Ho, D. T., Sweeney, C., & McGillis, W. R. Advances in quantifying air-sea gas exchange and environmental forcing, Annu. Rev. Mar. Sci., 1:213–44, 2009.

---

## Author Comment (AC2) · 1 Apr 2019

The presented study uses an atmospheric global climate model to determine the impacts of increased oceanic DMS emissions on the Arctic sulfate aerosol budget, cloud droplet number concentration (CDNC) and cloud radiative forcing. The main finding is that increased wet scavenging of sulfate in 2050 compensates for any increased aerosol production due to higher DMS emissions. Furthermore, significantly higher CDNC and more negative cloud radiative forcing are found in 2050 because of higher nucleation rates. The first half of the paper describes the comparison of a historical simulation (1991-2003) with observations from ship-based campaigns and at Alert in the Canadian Arctic. The second half of the paper reports results from a simulation experiment for 2000 and 2050. The illustrations and visualization of results adequately reflect the key findings of the analysis. The first half is rather well presented and written in a comprehendible way. The second half (Sect. 4) is very difficult to evaluate because of a lack of information and a confusing description of the simulations. The manuscript should have a section (for example after Sect. 2) that provides an overview of all details and motivation behind the simulations in one place. Moreover, it is unclear why the historical simulation was not used as present-day reference simulation for the future runs, but instead a new one, "2000", integrated over four years, is introduced on page 7. How can we be sure that "2000" has the same robustness as the historical simulation?

Response: First we would like to thank the reviewer for providing a detailed evaluation of our work which lead to an overall improvement in the documentation of the results. All referee comments are answered below following original comments. For the referee questions regarding different simulations for model evaluation and sensitivity analysis we provide explanations in the following two points:

1) We keep model evaluation section (i.e. Section 3) separate from sensitivity simulations (i.e. Section 4) in order to avoid confusion. The long-term simulations discussed in section 3 (i.e. "hisCon" and "hisNoDMS") were necessary in order to perform a comprehensive evaluation of model simulations of sulfate aerosol and contribution of DMS to total sulfate aerosol over global ocean regions. Comprehensive shipboard observations of sulfate aerosols over global ocean regions are available from 1991 to 2003, which we consider to be important for model evaluation. Therefore, in order to provide an estimate of DMS contribution to total sulfate aerosols, in hisNoDMS simulations DMS was set to zero over global oceans.

2) For the model sensitivity simulations, we focused on the Arctic region by changing DMS only in the region north of 60N. These simulations were run for four years using present-day conditions (i.e. 1998-2001) and potential future conditions (i.e. 2048-2051) with a total of 50 (four year) simulations. Due to such a large of number of model simulations, needed to better understand influence of different DMS scenarios under present-day and future conditions, it was necessary to limit the simulation time period to a total of 4 years for each of the ensemble members for present-day and future conditions. The simulation time period for present-day conditions overlaps with the time period of the hisCon and hisNoDMS simulations. Since we did not change any physical parameters or boundary conditions between model evaluation simulations (discussed in section 3) and the sensitivity simulations (discussed in section 4), the simulated aerosol concentrations are fully consistent with each other in the different types of simulations.

Scientifically, the study is questionable due to three major issues:

1) The future state of the atmosphere in 2050 is derived from integrated simulations of four years, 2048-2051, which seems to be too short. It appears that the climate simulation was meant to follow the RCP8.5 emission scenario. Have atmospheric concentrations of $CO_2$ and other trace gases as well as the global air temperatures in 2050 reached levels that are comparable to multi-decadal climate simulations for RCP8.5? In case this has been achieved during the 4-year run: has the output from the spin-up period been excluded from the analysis?

Response: For model boundary conditions for 2050 we used mean simulated sea ice and sea surface temperatures from a large 50-member ensemble which was conducted using the coupled version of the model (CanESM2), performed for CMIP5. Given the very large size of this ensemble, impacts of simulated natural variability on mean simulated sea ice and sea surface temperatures are negligible (Sigmond and Fyfe, 2016). The exact same greenhouse gas concentrations and emissions are specified for each individual ensemble member according to the RCP8.5 scenario. In order to further minimize the impact of natural variability in atmospheric and aerosol microphysical processes in simulations with CanAM, we use mean results from 5 different CanAM ensemble members with the exact same boundary conditions and baseline emissions. Ensemble members were generated by introducing random perturbations in radiative flux calculations which leads to small differences in meteorological conditions for each ensemble member. Similar to the approach used in comparable aerosol modelling studies using CMIP5 data (e.g. Ekman, 2014), our method ensures that ensemble mean results are robust and consistent with the boundary conditions and emissions that were used in the simulations. Please note the quantification of statistical uncertainty of the method in Figures 5-8. Based on these results we believe that we used a sufficiently large number of simulations in order to reliably address Arctic climate impacts of DMS emissions in our study.

Regarding the comment about spin-up period: We mention in the manuscript that the model spin-up years (1998 for 2000 and 2048 for 2050) were not included in the analysis.

2) Since the increase of CDNC due to increased emissions of DMS in 2050 compared to 2000 was significant, I would expect that the cloud microphysics are likewise significantly influenced by the higher number of cloud condensation nuclei (CCN), as this has been demonstrated for the summertime Arctic (Leaitch et al., 2013). If a cloud forms on a higher number of CCN the condensed water will be distributed over many small droplets rather than over a few large ones, given that the available amount of water is the same. An increase in CCN concentration results in faster evaporation rates owing to smaller cloud droplets. The faster evaporation rate leads to enhanced entrainment of sub-saturated air surrounding the cloud and a decrease in cloud fraction, in turn lowering the aerosol effect on cloud albedo (Zuidema et al., 2008). Have such aerosol-induced changes on the cloud macrophysics be considered?

Response: In grid cells that are affected by clouds, CanAM4.3 accounts for cloud albedo and lifetime effects (1st and 2nd aerosol indirect effects) as well the semi-direct effect. Parameterizations of droplet evaporation in the model do not account for aerosol effects, similar

to CMIP5 climate models. We agree that the representation of aerosol/cloud interactions is uncertain in climate models and would refer to the IPCC WG1 assessment for a summary of the scientific understanding of the impact of these processes on global climate. Although aerosol indirect effects are very difficult to constrain, some studies based on observations and cloud-resolving modelling indicate that cloud microphysical processes may produce negative or positive radiative forcings, depending on the meteorological situation and nature of the clouds (Stevens and Feingold, 2009). However, indirect effects in climate models are consistently associated with negative radiative forcings. A review of this topic is outside the scope of the current study.

3) Sea-salt particles and primary organic particles could be much more efficient CCN than particles derived from DMS (Quinn and Bates, 2011). If the sea salt or organic particle was already sufficiently large to serve as a CCN, the addition of DMS-derived sulfur to the particle will not increase the number of CCN. The increase of the primary sea-spray emissions with retreating ice, would not just be compensated by increased wet scavenging, but might outcompete DMS as precursor for CCNs. Clearly, increased emission of sea-salt aerosol will inhibit the development of precipitation. It will also cause more large CCNs, which can efficiently suppress activation of some of the smaller (sulfate) particles (O'Dowd et al. 1999).

Response: We agree that both sea salt and primary organic particles could be important sources of CCN. However, there is no consensus on the CCN activity of sea spray aerosols (including primary organic aerosols and sea salts) (Neukermans et al., 2018). Based on historical shipboard observations, Quinn et al. (2017) concluded that a small fraction of marine cloud condensation nuclei are made up of sea spray aerosol especially in regions north of 60N. Leaitch et al. (2016), based on recent observations in the Arctic region, also found that small particles (up to 20nm) are activated in summer. Similarly, Collins et al. (2017) reported frequent occurrence of activation of ultrafine particles in the Canadian Arctic Archipelago.

Specific Points:
- P.1 line 41: Mention that gaseous MSA also nucleates and plays an important role in the initial growth of new particles. Importantly, a recent study in the Canadian Arctic Archipelago by Willis et al. (2016) presents observational evidence that the growth of nucleation mode aerosol in the summertime Arctic is correlated with the presence of particulate MSA and organic species.

Response: Agreed, this is now mentioned in the text which reads as "*Willis et al. (2016) found that gaseous MSA may also play an important role in the initial growth of new particle formation.*" Further, please note that MSA is treated as sulfuric acid in model for simplicity, as indicated in the manuscript. Hence, we may be able to account for some of the impacts of MSA.

- P.2 line 71: DMS in water or in air. Suggest to denote seawater DMS as DMS(aq) and gaseous DMS as DMS(g).

Response: Thanks for noticing the omission. We changed the sentence to clarify that the reference is to surface seawater DMS concentrations here. We believe that our current notation

of referring to surface seawater DMS concentrations (instead of DMS(aq)) is sufficiently concise and we would prefer to keep this approach.

- P.3 line 98: How much of the produced DMSP is transferred to sediments? Does is not depend on grazing pressure how much DMS is actually produced? How sensitive are diatoms and haptophytes to seawater temperature changes?

Response: The questions posed here are well beyond the scope of our paper. We are well aware of the uncertainty surrounding estimates of future DMS emission in the Arctic, and this is why we tackle the problem using a sensitivity analysis with a wide range in sea-surface DMS concentrations (see next question). This being said, we will briefly reply the reviewers' questions below.

First of all, the reviewer should note that DMSP is not explicitly represented in our model. Rather, sea-surface DMS fields are prescribed from either a climatology of in situ measurements (Lana et al. 2011) or satellite-based estimates (Galí et al. 2018). The latter satellite algorithm does estimate DMSP from phytoplankton biomass, and subsequently DMS. As shown by Galí et al. (2015), this is possible because DMS concentrations adjust rapidly to changes in the plankton community, due to the short turnover time of DMS (1-2 days).

Regarding cell sinking: It is very unlikely that this process would significantly impact DMS(P) budgets due to the following two reasons. First, because high DMSP producers are generally small-celled and do not sink appreciably (except for colonial *Phaeocystis* blooms), i.e., they would sink at less than 1 m/d (in addition, they are motile!). In a mixed phytoplankton population, only diatom cells larger than about 30 μm would sink at appreciable speeds of >2 m d$^{-1}$, but diatoms generally are low DMSP producers and make a minor contribution to total DMSP stocks (McParland and Levine 2018). In consequence, DMSP turnover due to cell sinking out of the surface layer has a much longer turnover time than DMS production from DMSP degradation, which is typically 1-2 days (specific rates of 50% to 100% d$^{-1}$). Lizotte et al. (2008) found that DMSP turnover due to sinking was <2% d$^{-1}$ in a North Atlantic diatom bloom.

Regarding grazing pressure: again, this cannot be explicitly accounted for in our study. We refer the reviewer to the study of Galí et al. (2015), which showed how sea-surface DMS can be estimated from environmental variables (chiefly light, as done in the Galí et al. 2018 algorithm). In this approach, food-web interactions are not explicitly represented, but are partly accounted for in an implicit way through the light-mediated seasonal changes in the plankton community structure.

Regarding the last question: Future changes in the dominance of haptophytes vs. diatoms are difficult to predict. Yet, there is evidence for Atlantification of some Arctic sectors (Barents Sea) with northward propagation of coccolithophore (Emiliania hux.) blooms following polar front (e.g. Neukermans et al. 2018). (see next question).

- P.3 line 108: Please explain why a factor of 10 is used. Are there any projections about future DMS emission (from the same water column, not due to ice loss) that justify this order of magnitude increase?

Response: We did address this issue in detail in the introduction section of the manuscript. We understand that it is difficult to be certain about future Arctic seawater DMS concentrations. Some of the discussion that is included in the introduction section is given here:
"*Surface seawater DMS concentrations measured in the Canadian Arctic in July and August of 2014 and 2016 were substantially higher than those used by Lana et al. for July and August (e.g., NETCARE median concentrations of 4.4 nmol/L and 7.3 nmol/L, Martine Lizotte, personal communication; median concentration range from 0.5 to 4.4 nmol/L for Lana et al., https://saga.pmel.noaa.gov/dms/). Furthermore, melt ponds on sea ice represent a yet missing source of DMS in studies of the Arctic (Mungall et al., 2016; Ghahremaninezhad et al., 2016; Gourdal et al. 2018; Abbatt et al., 2018).*
*Long-term observational studies provide evidence that high DMSP-producing haptophytes are becoming more prevalent in the Arctic in the last decade (Winter et al., 2013; Nöthig et al., 2015; Soltwedel et al., 2016). Furthermore, Arrigo et al. (2008) suggest that primary productivity may increase more than 3 times compared to 1998-2002, if Arctic sea ice loss continues. A combination of a shift in the species composition and an increase in primary productivity (e.g. Yool et al., 2005; Vancoppenolle et al., 2013) could imply a multiplicative increase in surface seawater DMS concentrations in future climate.*"

Furthermore, projections based on extrapolation of satellite-based estimates suggest a 2-3 fold increase in Arctic DMS emission (north of 70N) for an ice free Arctic summer (May to August). This is quite uncertain and does not factor in changes in species distribution (Gali et al., 2019, submitted)."

We are not aware of any projections or other studies that would provide evidence for lower uncertainties in surface seawater DMS concentrations. In general, information about uncertainties in future Arctic surface seawater DMS concentrations is very limited in the available literature.

- P.4 line 127: How well does Piecewise Lognormal Approximation cope with newly introduced particles from nucleation?

Response: The model simulates binary homogeneous nucleation of sulfuric acid and water vapour. Newly formed particles grow by condensation and coagulation. The numerical treatment of these processes is highly accurate and compares well with other methods (von Salzen, 2006). Unfortunately, we are not aware of any measurements that would allow us to validate the representation of newly formed particles from nucleation in the Arctic. However, simulated concentrations of CN, CCN, and CDNC are realistic for the small number of available observations in the Arctic. We believe that further analysis is required in order to validate simulations of nucleation mode aerosol in the model, which is beyond the scope of this study.

- P.4 line 130: In high latitude regions - characterized by low temperatures and high wind speeds -estimated DMS transfer velocity from wind speed parameterizations will be biased high if only the Schmidt number normalization is used.

Response: Tesdal et al. (2016) considered the impact of different gas transfer velocity parameterizations and found that differences in these parameterizations lead to considerable uncertainties in global DMS fluxes. Based on the results of Tesdal et al. (2016), we selected a parameterization that seems to produce realistic fluxes of DMS in combination with the Lana et al. surface seawater DMS climatology. For instance, comparisons with observed sulfate concentrations in Fig. 1 and 2 produce reasonable agreement. Unfortunately, we are not aware of any additional measurements that we could use to directly validate the simulated DMS fluxes in the model.

We are not aware of any particular biases in transfer velocities in the Arctic. The Arctic summer seems to be characterized by relatively moderate to low wind speeds (see e.g. Hughes and Cassano, 2015) compared to other regions of the ocean. Regarding temperature, the Schmidt number (Sc) already includes temperature effects on gas diffusivity (Sc is defined as the ratio between the kinematic viscosity of seawater and DMS diffusivity) (Wanninkhof et al., 2009). Indeed, this results in low sea-air gas transfer coefficients in the Arctic, particularly in seasonally ice-covered waters which have low temperatures during most of the summer.

Finally, note also that the effects of ice shear on interfacial turbulence might also alter sea-air transfer k in the Arctic, causing departures from relationships based on wind speed. However, contradicting results have been reported, such that gas exchange was found to be either lower (Van der Loeff et al., 2014) or higher (Loose et al., 2014) than expected based on a linear scaling to percent ice cover. Therefore, it is reasonable to scale k by ice fraction.

- P.4 line 131: Does the model include oceanic emissions of sea-salt particles and primary organic particles? If not, that must be stated here.

Response: Yes, the model includes oceanic emissions of sea-salt particles but there are no emissions of organic aerosol species from the ocean. We added this information to the manuscript at the end of this paragraph.

- P.4 line 134 - 135: "MSA is treated as sulfuric acid in model for simplicity" – this is problematic in multiple ways. MSA forms in one step from the oxidation of DMS. Given it nucleates as sulfuric acid, then the atmospheric nucleation would be much too efficient. Please provide the average nucleation rates of Table 2 with literature data from the Arctic, e.g. Karl et al. (2012) and Leaitch et al. (2013). Although MSA is much less efficient than sulfuric acid in forming particles with water molecules, several laboratory studies and computational studies confirmed that MSA forms particles with alkylamines (Dawson et al., 2012; Chen et al., 2015; Xu et al., 2018). The presence of water seems to control the new particle formation in this system.

Response: The nucleation rates given in our Table 2, seem to be comparable with Karl et al. (2012, with values 0.04 to 0.1 cm$^{-3}$ s$^{-1}$), however we only have vertically integrated nucleation rates available from the model while Karl et al. reports near-surface values based on shipboard observations. We agree that the treatment of MSA is very simple. We currently don't have the capability to simulate MSA and adopted this approach in order to account for an enhancement in nucleation rates due to a combination of binary homogeneous nucleation of MSA and water vapour (inefficient) and new particle formation in the alkylamine/water vapour system. Alternatively, omitting MSA would likely lead to nucleation rates that are too low.

- P. 6, line 190 - 194: What explains the high modelled biogenic sulfate in June in Fig. 2c? Can it be related to the DMS seawater concentrations?

Response: There could be several explanations for this model bias. Generally, the model seems to overestimate biogenic source contributions for all months at this location, especially in June. This is consistent with biases at Alert according to Fig. 1.

We suspect that the biases may depend on the location of the comparison. Results in Fig. 1 indicate that sulfate concentrations agree better with observations in other regions of the ocean. As explained in the introduction and also shown by Tesdal et al. (2016), surface seawater DMS concentrations are particularly uncertain in the Arctic and therefore we cannot rule out biases in Arctic DMS emissions. In addition, the use of climatological oxidant concentrations may be problematic. Furthermore, deposition processes are uncertain in the Arctic which leads to large differences in simulated Arctic aerosol concentrations in different models (e.g. Mahmood et al., 2016). Further investigations are needed in order to understand the causes of the biases, which is beyond the scope of this paper.

- P. 6, line 205: Please explain the occurrence of the October peak in the annual cycle.
Response: We are not aware of an explanation from earlier studies. Sharma et al. (2012) showed that MSA concentrations at Alert are anti-correlated with sea ice fraction.  It is possible that the peak in October is related to increased fluxes of DMS into the Arctic atmosphere due to the minimum in sea ice fraction in September. It is also possible that DMS is transported to Alert from lower (subpolar) latitudes, where fall phytoplankton blooms are a dominant feature of the marine ecosystem (Ardyna et al., 2014). In any event, the focus of the study is on annual mean results and we have not investigated this.

- P. 6, line 217: Please provide more details about the radiative flux calculations in Sect. 2. Which parameters were perturbed? A table with parameters and perturbation values for all 25 simulations would be very helpful.

Response: Perturbations to model variables are used in order to introduce random variability in climate model results in order to assess natural atmospheric variability in model ensembles. Different techniques are used by the climate modelling community. Lifetimes of frontal system

and, more generally, time scales associated with conversion of heat and moisture in the atmosphere are typically much shorter than the model spinup time period. Consequently, any perturbation to initial conditions in the atmosphere, whether radiative fluxes, winds or temperatures are perturbed, will produce statistically indistinguishable variations in meteorological variables at the end of the model spin up time period. This is a consequence of the highly non-linear atmospheric system, which is well documented in the literature. A review of ensemble modelling techniques and details of the well-established perturbation method employed in our model does not seem appropriate to us in the context of our study.

- P. 7, line 241: Explain better what "corresponding simulation" means here.

Response: We have tried elaborating on this in the manuscript so that the relevant sentence now reads: "*Simulated horizontal wind and temperature in each individual member of an ensemble (i.e. 5 separate simulations) were nudged towards specified results from a corresponding simulation (i.e. separate free running model simulation) with CanAM4.3 using a nudging time scale of 6 h.*"

- P. 7, line 242-244: An explanation is missing here, why a new simulation "2000" and not the historical simulation "hisCont" was used as present-day reference.
Response: This has been answered in detail as response to the general comment above.

- P. 8 line 286: There could be two reasons why wet deposition increased, growth of particles to larger sizes by condensation of DMS oxidation products which makes them more accessible to wet scavenging or the increase of precipitation rates. Please provide information on the average precipitation rates in the simulations in Table 2. Also give the average liquid water content of clouds in Table 2.

Response: We did provide information on several parameters including precipitation, wet deposition and liquid water content of clouds in supplementary materials (see Tables S1-S3 and Figures S1-S2).

- P. 8 line 295: Is "10ˆ7 mˆ-3" the number change or the absolute number of CDNC? Please set the value in relation to average CDNC in the present-day simulation.

Response: It represents maximum change in CDNC relevant to CNTRL run which is important to explain quantitative changes shown in Figure 6. Average CDNC in relation to control runs is already provided in Table 2 both for present-day and future simulations.

- P. 11 line 375: How much does in-cloud sulfate production increase? More details on aqueous-phase production of sulfate in the model and about treatment of in-cloud scavenging should be provided in Sect. 2.

Response: Information about in-cloud sulfate production was provided in supplementary materials (please see Fig. S3). For details regarding parameterizations of aerosol and cloud

interactions in the model please see von Salzen (2006) and von Salzen et al., (2013), which are also cited in the paper.

- P. 11 line 390 - 395: Could you elaborate on the expected feedbacks due to increased SST? Warmer water would be less favorable for diatoms but the solubility of DMS would be lower.

Response: Lower DMS solubility enhances its flux to the atmosphere. However, this has a marginal effect on seawater DMS concentrations because the latter are set by the dynamic equilibrium between sources and sinks, and ventilation is generally <10% of DMS sinks in the upper mixed layer of the ocean (e.g. Galí et al. 2015). Therefore, ventilation generally does not control sea-surface DMS concentrations.

- P. 11 line 396-397: Steady-state atmospheric oxidant concentrations are not the only additional uncertainty. The assumptions on nucleation rates and in-cloud scavenging of sulfate seem to be critical for the conclusion of this study.

Response: We agree, the final paragraph of the conclusion section is modified and now reads as: "*The model simulations used in the current study are not interactively coupled with ocean and sea ice DMS and therefore rely on specified surface seawater DMS concentrations. The current model version does not account for increases in sulfuric acid condensation sink due to increased emissions of sea salt and organic aerosols from the open ocean, which may lead to overestimates in nucleation rates in the simulations. Additional uncertainty in the strength of the feedback arises from the fact that atmospheric oxidant concentrations are assumed to be steady in our study. More comprehensive assessments of the strength and impacts of DMS/climate feedbacks in the Arctic will become possible once a new generation of Earth System Models with interactive ocean and sea ice DMS, chemistry, and climate processes becomes available.*"

References:

Abbatt, J. P. D., Leaitch, W. R., Aliabadi, A. A., et al.: New insights into aerosol and climate in the Arctic, Atmos. Chem. Phys., https://doi.org/10.5194/acp-2018-995, 2018.

Arrigo, K. R., van Dijken G., and Pabi S.: Impact of a shrinking Arctic ice cover on marine primary production, Geophys. Res. Lett., 35, L19603, doi: 10.1029/2008GL035028, 2008.

Collins, D. B., Burkart, J., Chang, R. Y.-W., Lizotte, M., Boivin-Rioux, A., Blais, M., Mungall, E. L., Boyer, M., Irish, V. E., Massé, G., Kunkel, D., Tremblay, J.-É., Papakyriakou, T., Bertram, A. K., Bozem, H., Gosselin, M., Levasseur, M., and Abbatt, J. P. D.: Frequent ultrafine particle formation and growth in Canadian Arctic marine and coastal environments, Atmos. Chem. Phys., 17, 13119-13138, https://doi.org/10.5194/acp-17-13119-2017, 2017

Ekman, A. M. L. (2014), Do sophisticated parameterizations of aerosol-cloud interactions in CMIP5 models improve the representation of recent observed temperature trends?, J. Geophys. Res. Atmos., 119, 817–832, doi:10.1002/2013JD020511.

Galí, M., and Simó, R. A meta-analysis of oceanic DMS and DMSP cycling processes: Disentangling the summer paradox. Global Biogeochemical Cycles 29 (4), 496-515, 2015.

Galí, M., Devred, M., Babin, M., and Levasseur, M. Decadal increase in Arctic dimethylsulfide emission. *Submitted*.

Galí, M., Levasseur, M., Devred, E., Simó, R., and Babin, M.: Sea-surface dimethylsulfide (DMS) concentration from satellite data at global and regional scales, Biogeosciences, https://doi.org/10.5194/bg-2018-18, 2018.

Ghahremaninezhad, R., Norman, A.-L., Abbatt, J. P. D., Levasseur, M., and Thomas, J. L.: Biogenic, anthropogenic and sea salt sulfate size-segregated aerosols in the Arctic summer, Atmos. Chem. Phys., 16, 5191–5202, https://doi.org/10.5194/acp-16-5191-2016, 2016.

Gourdal, M., Lizotte, M., Massé, G., Gosselin, M., Poulin, M., Scarratt, M., Charette, J., and Levasseur, M.: Dimethyl sulfide dynamics in first-year sea ice melt ponds in the Canadian Arctic Archipelago, Biogeosciences, 15, 3169-3188, https://doi.org/10.5194/bg-15-3169-2018, 2018.

Hughes, M., and J. J. Cassano (2015), The climatological distribution of extreme Arctic winds and implications for ocean and sea ice processes, J. Geophys. Res. Atmos., 120, 7358–7377, doi:10.1002/2015JD023189.

Karl, M., Leck, C., Gross, A., and Pirjola, L.: A study of new particle formation in the marine boundary layer over the central Arctic Ocean using a flexible multicomponent aerosol dynamic model, Tellus B, 64, 17158, doi:10.3402/tellusb.v64i0.17158, 2012.

Lana, A., Bell, T. G., Simó, R., Vallina, S. M., Ballabrera-Poy, J., Kettle, A. J., Dachs, J., Bopp, L., Saltzman, E. S., Stefels, J., Johnson, J. E., and Liss, P. S.: An updated climatology of surface dimethlysulfide concentrations and emission fluxes in the global ocean, Global Biogeochem. Cycles, 25, GB1004, doi:10.1029/2010GB003850, 2011.

Leaitch, W. R., Korolev, A., Aliabadi, A. A., Burkart, J., Willis, M. D., Abbatt, J. P. D., Bozem, H., Hoor, P., Köllner, F., Schneider, J., Herber, A., Konrad, C., and Brauner, R.: Effects of 20–100 nm particles on liquid clouds in the clean summertime Arctic, Atmos. Chem. Phys., 16, 11107-11124, https://doi.org/10.5194/acp-16-11107-2016, 2016.

Lizotte, M., Levasseur, M., Scarratt, M. G., Michaud, S., Merzouk, A., Gosselin, M., and Pommier, J.: Fate of dimethylsulfoniopropionate (DMSP) during the decline of the northwest Atlantic Ocean spring diatom bloom, Aquat. Microb. Ecol., 52, 159–173, 2008.

Loose, B., McGillis, W. R., Perovich, D., Zappa, C. J., & Schlosser, P. (2014). A parameter model of gas exchange for the seasonal sea ice zone. Ocean Science, 10(1), 17-28.

McParland, E. L., & Levine, N. M. (2019). The role of differential DMSP production and community composition in predicting variability of global surface DMSP concentrations. *Limnology and Oceanography*, *64*(2), 757-773.

Mungall, E. L., Croft, B., Lizotte, M., Thomas, J. L., Murphy, J. G., Levasseur, M., Martin, R. V., Wentzell, J. J. B., Liggio, J., and Abbatt, J. P. D.: Dimethyl sulfide in the summertime Arctic atmosphere: measurements and source sensitivity simulations, Atmos. Chem. Phys., 16, 6665–6680, https://doi.org/10.5194/acp-16-6665-2016, 2016.

Neukermans, Harmel, Gali et al. 2018. Harnessing remote sensing to address critical science questions on ocean-atmosphere interactions. Elem Sci Anth, 6: 71., DOI:https://doi.org/10.1525/elementa.331

Nöthig, E.-M., Bracher, A., Engel, A., Metfies, K., Niehoff, B., Peeken, I., Bauerfeind, E., Cherkasheva, A., Gäbler-Schwarz, S., Hardge, K., Kilias, E., Kraft, A., MebrahtomKidane, Y., Lalande, C., Piontek, J., Thomisch, K., and Wurst, M.: Summertime plankton ecology in Fram Strait—a compilation of long- and short-term observations, Polar Research, 34:1, DOI: 10.3402/polar.v34.23349, 2015.

Quinn, PK, Coffman, DJ, Johnson, JE, Upchurch, LM and Bates, TS. 2017. Small fraction of marine cloud condensation nuclei made up of sea spray aerosol. *Nat Geosci* 10: 674–679. DOI: https://doi. org/10.1038/ngeo3003.

Sharma, S., et al. (2012), Influence of transport and ocean ice extent on biogenic aerosol sulfur in the Arctic atmosphere, J. Geophys. Res., 117, D12209, doi:10.1029/2011JD017074.

Sigmond, M., and Fyfe, J. C.: Tropical Pacific impacts on cooling North American winters, Nat. Clim. Change, 6, 970-974, doi:10.1038/nclimate3069, 2016.

Soltwedel, T., Bauerfeind, E., Bergmann, M., Bracher, A., Budaeva, N., Busch, K., Cherkasheva, A., Fahl, K., Grzelak, K., Hasemann, C., Jacob, M., Kraft, A., Lalande, C., Metfies, K., Nöthig, E.-M., Meyer, K., Quéric, N.-V., Schewe, I., Włodarska-Kowalczuk M., Klages, M.: Natural variability or anthropogenically-induced variation? Insights from 15 years of multidisciplinary observations at the arctic marine LTER site HAUSGARTEN, Ecological Indicators, 65, 89-102, https://doi.org/10.1016/j.ecolind.2015.10.001, 2016.

Stevens, B., and G. Feingold (2009), Untangling aerosol effects on clouds and precipitation in a buffered system, Nature, 461(7264), 607–613, doi:10.1038/nature08281.

Tesdal, J.-E., Christian, J. R., Monahan, A. H., and von Salzen, K.: Evaluation of diverse approaches for estimating sea-surface DMS concentration and air-sea exchange at global scale, Environmental Chemistry, doi: 10.1071/EN14255, 2016.

van der Loeff, M. M. R., Cassar, N., Nicolaus, M., Rabe, B., & Stimac, I. (2014). The influence of sea ice cover on air-sea gas exchange estimated with radon-222 profiles. Journal of Geophysical Research: Oceans, 119(5), 2735-2751.

Vancoppenolle, M., Bopp L. , Madec G., Dunne J. , Ilyina T., Halloran P. R. , and Steiner N.: Future Arctic Ocean primary productivity from CMIP5 simulations: Uncertain outcome, but consistent mechanisms, Global Biogeochem. Cycles, 27, 605–619, doi:10.1002/gbc.20055, 2013.

von Salzen, K., Scinocca, J. F., McFarlane, N. A., Li, J., Cole, J. N. S., Plummer, D., Verseghy, D., Reader, M. C., Ma, X., Lazare, M., and Solheim, L.: The Canadian Fourth Generation Atmospheric Global Climate Model (CanAM4). Part I: Representation of physical processes, Atmos. Ocean, 51, 104–125, doi:10.1080/07055900.2012.755610, 2013.

von Salzen, K.: Piecewise log-normal approximation of size distributions for aerosol modelling, Atmos. Chem. Phys., 6, 1351–1372, 2006.

Willis, M. D., Burkart, J., Thomas, J. L., Köllner, F., Schneider, J., Bozem, H., Hoor, P. M., Aliabadi, A. A., Schulz, H., Herber, A. B., Leaitch, W. R., and Abbatt, J. P. D.: Growth of nucleation mode particles in the summertime Arctic: a case study, Atmos. Chem. Phys., 16, 7663–7679, doi:10.5194/acp-16-7663-2016, 2016.

Winter, A., Henderiks, J., Beaufort, L., Rickaby, R. E. M., and Brown, C. W.: Poleward expansion of the coccolithophore Emiliania huxleyi, Journal of Plankton Research, 36, 316–325, 2014.

Yool, A., Popova, E. E., and Coward, A. C.: Future change in ocean productivity: Is the Arctic the new Atlantic?, J. Geophys. Res. Oceans, 120, 7771-7790, doi:10.1002/2015JC011167, 2015.